# Rift thermal inheritance in the SW Alps (France): insights from RSCM thermometry and 1D thermal numerical modelling

Naïm Célini [1,2,3], Frédéric Mouthereau[2], Abdeltif Lahfid[3,4], Claude Gout[1], Jean-Paul Callot[1]

[1] Université de Pau et des Pays de l'Adour, E2S UPPA, CNRS, TotalEnergies, LFCR, Pau, France.

[2] Géosciences Environnement Toulouse, Université de Toulouse Paul Sabatier, CNRS, IRD, Toulouse, France.

[3] BRGM, F-45060 Orléans, France.

[4] ISTO, UMR7327, Université d'Orléans, CNRS, BRGM, F-45071 Orléans, France.

*Correspondence to: celini.naim@gmail.com*

**ABSTRACT**

Conceptual models of orogenic accretionary prisms assume that peak temperatures ($T_{max}$) increase towards the internal domains as crustal rocks are accreted from the lower to the upper plate. Yet, the recognition of pre-orogenic heating events in mountain belts questions the magnitude of thermal overprint during nappe stacking. Using Raman Spectroscopy on Carbonaceous Material (RSCM) to calculate $T_{max}$, we have investigated the thermal record of Lower Jurassic to Eocene strata exposed along six stratigraphic sections at the front of the Digne Nappe (SW Alps), from the Devoluy Massif to the Castellane Arc. Our results highlight two groups of depth-dependent temperatures: (1) a regionally extensive and constant $T_{max}$ up to 300-330°C measured in the Jurassic succession and (2) regionally variable lower temperatures (<150°C) recorded either in the upper Mesozoic or the syn-orogenic sequence. Modelling shows that the highest paleotemperatures were achieved during the Early Cretaceous (~130 Ma), associated with the Valaisan-Vocontian rifting, while the lowest $T_{max}$ reflect post-rift thermal relaxation in the Alpine foreland basin. This study provides a striking new example where mid-crustal paleotemperatures measured in sediments accreted from the downgoing plate are inherited. An estimated peak thermal gradient of 80-90°C/km requires crustal thickness of 8-10 km during the Early Cretaceous, hence placing constraints for tectonic reconstruction of rift domains and geophysical interpretation of current crustal thickness in the SW Alps. These results call for the careful interpretation of paleothermal data when they are used to identify past collisional thermal events. Where details of basin evolution are lacking, high-temperature record may be misinterpreted as syn-orogenic, which can in turn lead to an overestimation of both orogenic thickening and horizontal displacement in mountain belts.

**INTRODUCTION**

The identification of collisional events through Earth history lies in our capacity to distinguish in the rock record and crustal structure the effect of crustal thickening among other tectono-magmatic events unrelated to plate convergence. Conceptual models of orogenic accretionary prisms describe the increase of peak temperatures ($T_{max}$) from the external fold-and-thrust belts to the internal high-grade metamorphic domains as a result of tectonic burial heating and thermal advection due to nappe stacking and erosion as crustal rocks are accreted (Barr et al., 1991). Therefore, a potential relevant indicator of collision is the onset of heating related to crustal thickening of previously subducted rocks (e.g. Soret et al., 2021). However, it has been shown that mid-crustal temperatures (i.e. at least 300°C) reconstructed in accreted sedimentary rocks can represent rift-related thermal peaks not overprinted by crustal thickening. This has been substantiated in low-convergence collisional orogens such as the Pyrenees (Izquierdo-Llavall et al., 2020; Vacherat et al., 2014; Clerc and Lagabrielle, 2014; Ducoux et al., 2021) and the Iberian Range (Rat et al., 2019) as well as in higher convergence collisional orogens such as Taiwan (Conand et al., 2020). In the Alps, the preservation of pre-Alpine contacts in high-pressure metamorphic units of formed during the development of the European margin of the Alpine Tethys (e.g. Beltrando et al., 2014) and the current debates regarding the interpretation of pressure-temperature-time paths (Pleuger and Podladchikov, 2014; Luisier et al., 2019; Schmalholz and Podladchikov, 2013) have raised questions about the magnitude of continental subduction and the role of pre-collision evolution on the orogeny dynamics. Therefore, we need to improve quantification of pre- versus syn-orogenic thermal evolution in orogenic wedges to reduce uncertainties on tectonic reconstructions and recognition of past collisional events.

The Western Alps are an archetypal mountain belt formed by the crustal accretion of the European and Adria paleomargins that followed the subduction of the Alpine Tethys (e.g. Handy et al., 2010). $T_{max}$ of about 300°C reported from Raman Spectroscopy on Carbonaceous Material (RSCM) and mineral thermometry in accreted sediments of the European plate (e.g. Briançonnais and Schistes Lustrés units; Fig. 1) are commonly attributed to syn-convergence thermal overprint in the internal units (e.g. Gabalda et al., 2009; Lanari et al., 2012). RSCM temperatures estimated in pre-collision Mesozoic sediments of the external units are also close to or above 300°C and have been interpreted as reflecting burial of the European margin beneath the internal units of the belt (Deville and Sassi, 2006; Bellanger et al., 2015). However, these mid-crustal temperatures are

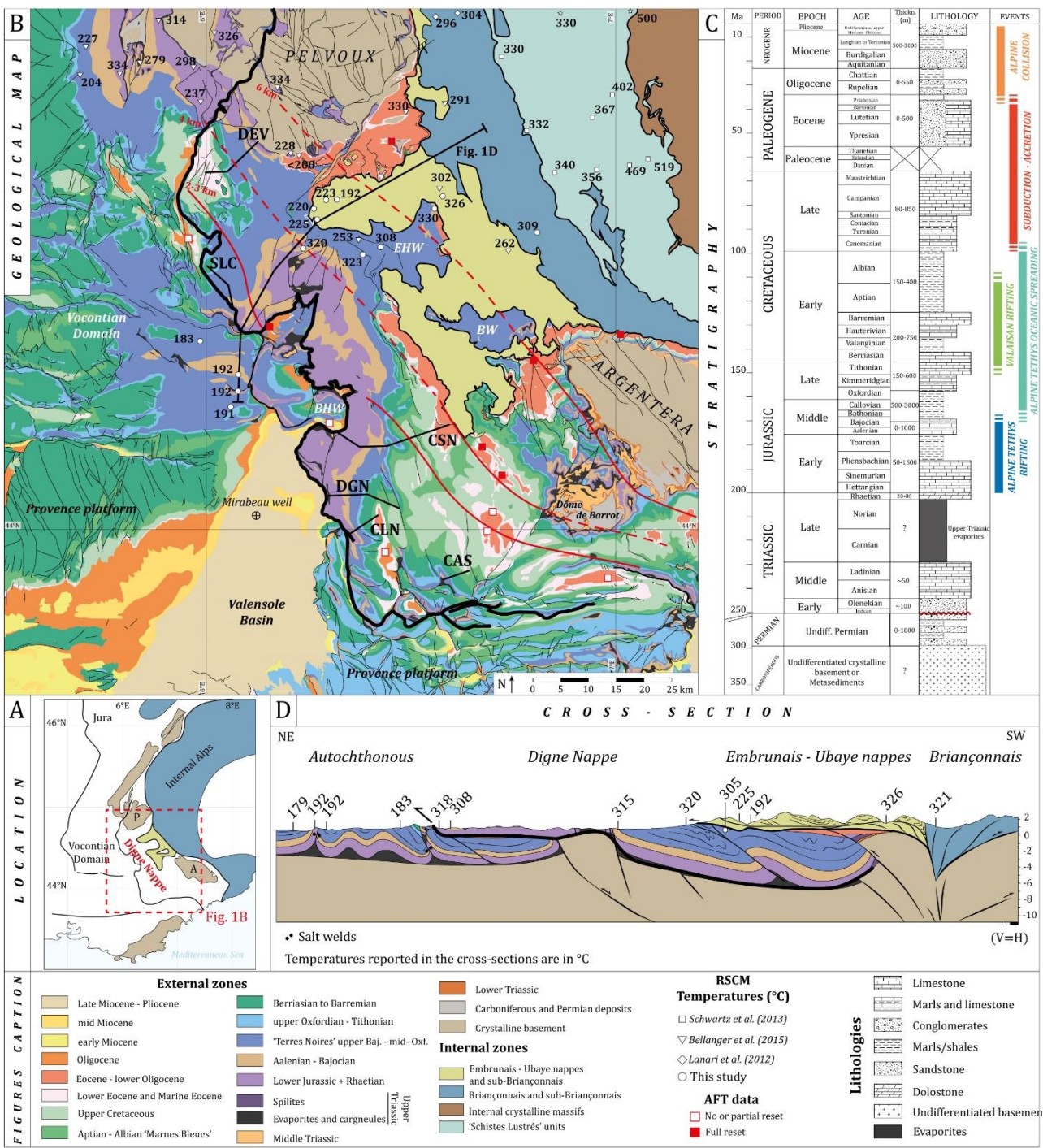

Figure 1: (A) Location of the study area in the regional framework of the Western Alps. (B) Simplified geological map of the SW Alps with the position of the sampled stratigraphic sections, 15 original RSCM temperatures and RSCM temperatures from the literature (Schwartz et al., 2013; Lanari et al., 2012), modified from the BRGM geological maps at 1/250000. The sections are from NNW to SSE: DEV = Devoluy; SLC = La Saulce; CSN = Cousson; DGN = Digne; CLN = Coulnier and CAS = Castellane. Red lines are reconstructed isopachs of Cenozoic foreland deposits based on published apatite fission-track data (Labaume et al., 2008; Jourdan et al., 2013; Schwartz et al., 2017). BHW=Barles Half Window; BW=Barcelonnette Window and EHW=Embrun Half Window. (C) Simplified synthetic stratigraphic column of the area, modified from Célini et al. (2022). (D) Cross-section through the northern part of the Digne Nappe and the Embrunais-Ubaye nappes containing temperatures from this study. See location in (B).

at odds with the expected decrease of tectonic burial in the external parts of orogenic accretionary wedges and

may instead reflect a thermal event prior to collision, a possibility that is yet to be assessed.

Here, we focus on the Digne Nappe, characterised by a thick Mesozoic-Cenozoic section transported

above the Digne thrust, at the front of the external SW Alps (e.g. Gidon, 1997). The low apparent internal

deformation makes this region suitable for deciphering the thermal impact of rifting events from other tectonic

phases. This study combines 90 new RSCM measurements from samples collected at the front of the Digne

Nappe along continuous stratigraphic successions that recorded, prior to the onset of nappe emplacement, the

evolution of the European margin from the opening of the Alpine Tethys until the early stages of the Alpine

foreland. To resolve the timing of pre-orogenic versus syn-orogenic thermal imprints, a set of modelled time-

temperature scenarios has been designed. Our results lead to the re-evaluation of the importance of rifting-

related, lower plate thermal inheritance, and provide constraints on the palaeogeography and rift architecture in

the SW Alps.

**GEOLOGICAL SETTING**

The SW Alps comprise five tectono-stratigraphic domains (Fig. 1). From NE to SW they are: (1) the

Internal Crystalline Massifs that represent the crust of the stretched European margin subducted and rapidly

exhumed in Eocene times; (2) the Liguro-Piemont domain, made of Jurassic to Paleocene meta-sediments of

the Schistes Lustrés complex that originated from the oceanic floor and continent-ocean transition of the Alpine

Tethys; (3) the continental block of the Briançonnais and sub-Briançonnais domains, made of Paleozoic,

Mesozoic and Cenozoic meta-sediments characterised by limited crustal thinning during the Alpine Tethys

opening; (4) the Upper Cretaceous – Paleocene Helminthoid flyschs of the Embrunais-Ubaye nappes with slices

of Briançonnais and sub-Briançonnais units at their base, and (5) the Dauphinois/Helvetic domain, which forms

the external Alps, to the west of the Penninic front, and characterised by the External Crystalline Massifs and

by Paleozoic, Mesozoic and Cenozoic sediments constituting the external fold-and-thrust belt (Fig. 1B).

In the Western Alps, kinematic reconstructions of the Alpine Tethys involve two different oceanic

domains and/or hyper-extended margins separated by the Briançonnais continental block: the Alpine Tethys

sensu stricto and the Valaisan Domain (e.g. Stampfli and Borel, 2002). In the East, the Alpine Tethys opened

during the Middle to Late Jurassic as attested by the ages of mafic rocks ranging from 166 to 155 Ma (e.g.

Manatschal and Müntener, 2009). The continental break-up resulted in the formation of the slow-spreading

oceanic domain of the Alpine Tethys oriented NE-SW between the European and Adria margins (Mohn et al.,

2014; Graciansky et al., 1989; Lemoine et al., 1986; Manatschal et al., 2021). In the West, the nature and the

age of stretching of the Valaisan Domain, located between the Briançonnais and the Penninic front is more controversial. Ages span from (1) the Late Jurassic based on crystallisation age of a metagabbro (161 Ma; Liati et al., 2005) and indirect evidence of pre-Cretaceous mantle exhumation, to (2) the Early Cretaceous (130-90) as suggested by U/Pb dating of metamorphic zircon U-rich rims (Beltrando et al., 2007) and the interpretation of the Barremian-Aptian succession as post-rift deposits synchronous with the onset of oceanic spreading (Loprieno et al., 2011). These two contrasting views do not contradict evidence that the initiation of rifting occurred during the Early Jurassic in the Dauphinois Domain (e.g. Lemoine et al., 1986), leading to the development of a necking zone since the Pliensbachian in the External Crystalline Massifs (Mohn et al., 2014; Ribes et al., 2020; Dall'Asta et al., 2022). Thus, we consider the formation of the European margin to be a long-lasting process. Rifting started in the Early Jurassic and then evolved from the Late Jurassic into a hyper-extended margin and slow-spreading ridge in the Alpine Tethys. During the Early Cretaceous, a distinct thermal/metamorphic and magmatic event occurred in the Valaisan Domain (Fig. 1C). The Early Cretaceous events appears synchronous with the renewed extension in the so-called Vocontian Domain (Fig. 1A and 1B), the SW Alps and the Provence platform (Fig. 1B) (Graciansky and Lemoine, 1988; Hibsch et al., 1992; Jourdon et al., 2014; Angrand and Mouthereau, 2021).

Oceanic spreading lasted until the Late Cretaceous when the subduction of the eastern part of the Alpine Tethys started (Handy et al., 2010; Coward and Dietrich, 1989). Convergence in the Western Alps is characterised by two major stages (e.g. Ford et al., 2006). The first one, which occurred during the Eocene (55-34 Ma) reflects a period of subduction and rapid exhumation as the Alpine accretionary prism moved to the N-NW (Dumont et al., 2012, 2011; Merle and Brun, 1984; Lanari et al., 2014). The second stage occurred in the Late Eocene-Early Oligocene (34-30 Ma). It corresponds to the onset of widespread crustal thickening and collision in the Western Alps. This event is characterised by the onset of W- to SW-oriented shortening in the southern Western Alps resulting in the deposition of Alpine foreland deposits during the Priabonian-Rupelian (Bellahsen et al., 2014; Dumont et al., 2011; Fry, 1989; Coward and Dietrich, 1989; Simon-Labric et al., 2009).

Structuring the foreland between the variscan Pelvoux and Argentera crystalline massifs, the Digne main thrust front is oriented NNW-SSE and dies out to the north near the western border of the Pelvoux. It turns to an E-W direction in the Castellane Arc to the south (Fig. 1B). To the east, the units of the Digne Nappe are overthrusted by the Embrunais-Ubaye (E-U) nappes emplaced at around 30 Ma, during and shortly after the deposition of the Eocene-Oligocene foreland deposits (Kerckhove and Thouvenot, 2010).

The Digne Nappe is made of Triassic to Cretaceous sediments that consist mainly of alternating limestones and marls (Fig. 1C) and Eocene-Oligocene foreland remnants made of Nummulitic limestones, Globigerina marl and continental sandstones and conglomerates (Fig. 1C) transported towards the SW over 20-25 km (Gidon, 1997; Faucher et al., 1988; Lickorish and Ford, 1998). These successions belong to the former European passive margin (Gidon, 1997; Baudrimont and Dubois, 1977; Debrand-Passard et al., 1984) and are affected by salt tectonics since the Early Jurassic including the syn-orogenic phase (e.g. Célini et al., 2020, 2021, 2022). As the margin was inverted during the Miocene, Triassic evaporites provided an efficient décollement level for the Digne Nappe (Gidon and Pairis, 1985; Gidon, 1997; Faucher et al., 1988; Lickorish and Ford, 1998). The latest thick-skinned phase of deformation of the Digne Nappe, which is detected on seismic tomographic images as a major crustal-scale thrust (Nouibat et al., 2022), occurred during the Mio-Pliocene since ca. 10 Ma (Ford et al., 2006; Schwartz et al., 2017).

## SAMPLES AND METHOD

### RSCM thermometry along reconstructed vertical sections

The RSCM approach is based on the temperature-dependant and irreversible natures of Carbonaceous Material (CM) maturation (e.g. Henry et al., 2019). The Raman spectra of CM change with increasing temperature so a given spectrum thus reveals the $T_{max}$ experienced by a CM-rich rock through its history (Fig. 2). Several calibrations for the use of the RSCM thermometry, corresponding to various ranges of temperatures and geological contexts, were published (e.g. Beyssac et al., 2002; Rahl et al., 2005; Aoya et al., 2010; Lahfid et al., 2010; Kouketsu et al., 2014; Schito et al., 2017). The RSCM approach has been successfully applied to calculate $T_{max}$ in organic-rich rocks and infer the tectono-thermal histories of mountain belts and sedimentary basins (e.g. Gabalda et al., 2009; Conand et al., 2020; Ducoux et al., 2021).

For this study, the RSCM analyses have been performed at the BRGM (Orléans, France) using a Renishaw inVia Reflex microspectrometer with a diode-pumped solid-state laser source excitation of 514.5 nm. The laser power that reached the surface of the sample, using a Leica DM2500 microscope and a x100 objective (numerical aperture = 0.90), never exceeded 0.1 mW. For each day of measurement, the microspectrometer was calibrated with a 520.5 cm$^{-1}$ line of an internal silicon standard. The Rayleigh diffusion was eliminated by edge filters, and the Raman signal dispersed onto an 1800 lines/mm monochromator before analysis by deep depletion

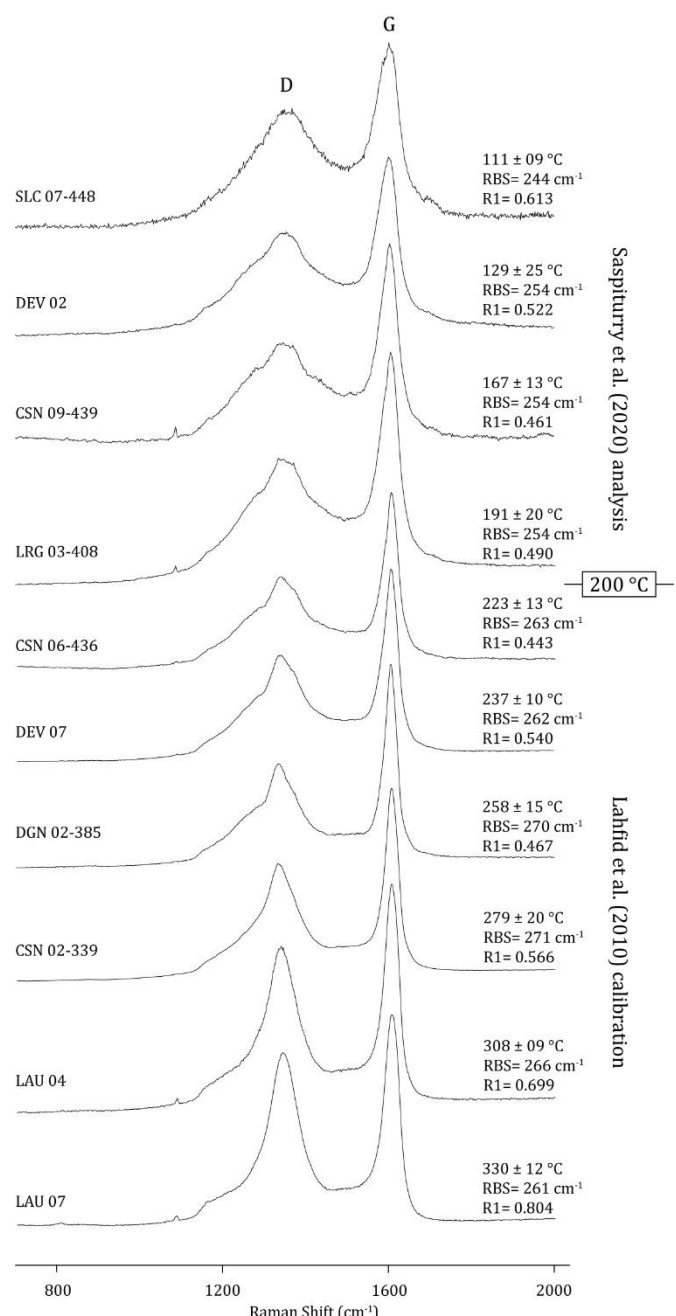

*Figure 2: Representative Raman spectra of carbonaceous materials for a selection samples and the corresponding temperatures obtained using Lahfid et al. (2010) calibration and Saspiturry et al. (2020) quantitative analysis. The RBS is the Raman Band Separation which represents the distance between the D and G bands. R1 is the Raman parameter of the intensity ratio between the D band and the G band. See also Suppl.Table for details regarding the RBS and R1.*

CCD detector (1024 x 256 pixels). For each sample, 10 to 15 measurements were acquired for data consistency, whenever the quality of the sample allowed it.

We selected six 2.5 to 5 km-thick stratigraphic sections named DEV (Devoluy), SLC (La Saulce), CSN (Cousson), DGN (Digne), CLN (Coulnier), and CAS (Castellane) (Fig. 3) distributed from NNW to SSE over a distance of 80 km along the Digne thrust front (Fig. 1B). A total of 63 samples have been retrieved from pre-collisional carbonate deposits of Early Jurassic to Late Cretaceous ages, and two in Eocene foreland deposits

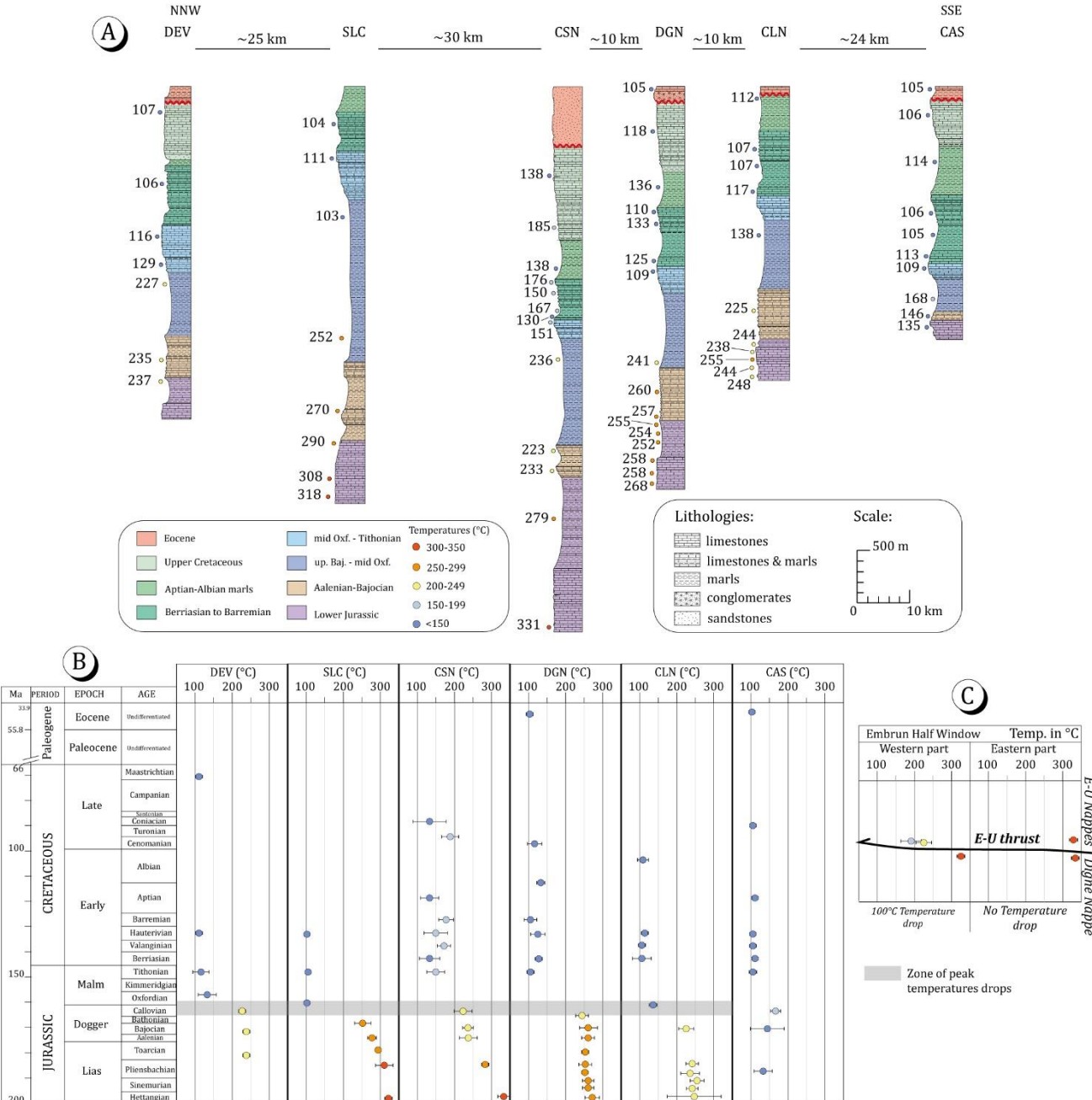

*Figure 3: (A) Stratigraphic sections along the front of the Digne Nappe with the RSCM-derived peak temperatures. See Data Repository 1 for details about the samples and Fig. 1 for the location of the sections. (B) Evolution of the RSCM temperatures through geological time scale along the sedimentary columns and (C) in two locations of the Embrun Half Window.*

(CAS and DGN, Fig. 3A). The sampled sections are assumed to represent the original vertical stratigraphic sections in the Alpine foreland basin prior to nappe emplacement in the Mio-Pliocene. This dataset was complemented by 15 RSCM analyses in the autochthonous units, in the E-U nappes and the Embrun half window (Fig. 1B and 1D). We applied the RSCM calibration of Lahfid et al. (2010) for temperatures ranging between 200 and 340°C, and added the qualitative analysis of Saspiturry et al. (2020) for lower temperatures between 100 and 200°C (Fig. 2).

**Numerical modelling with basin model**

To quantitatively examine which tectonic scenario best reproduces the distinct groups of depth-dependent temperatures observed along the Digne thrust front, we used 1D numerical modelling TemisFlow® (Doligez et al., 1986). This basin modelling code simulates, among other processes, the compaction of the porous medium and heat transfer during a basin history. These physical phenomena are modelled by a set of partial differential equations representing the temperature equation, the mass balances of the solid and the water phase, coupled with a compaction law (Schneider et al., 2000). Heat transport uses as input: basal heat flow computed from a time evolutive 3 layers lithosphere model, surface paleotemperatures, thermal conductivity of formations in the stratigraphic column and burial history, including erosion events. Both evolution of the crustal thicknesses linked to rifting events and erosion thicknesses were the parameters we used to match the $T_{max}$ data.

Three scenarios have been tested:

(1) the temperatures are only due to burial throughout the Mesozoic and Cenozoic without additional heat induced by lithospheric thinning (No Rift-model);

(2) the temperatures reflect the advection of heat during crustal and mantle thinning that resulted from the Alpine Tethys rifting event in the Early-Middle Jurassic (201-169 Ma), followed by a period of cooling after 125 Ma caused by thermal relaxation, i.e. the Lithosphere-Asthenosphere boundary (LAB) deepens, and then the burial in the Alpine foreland during the Cenozoic (One Rift-model);

(3) the temperatures are caused by two rifting events during the Early-Middle Jurassic (Alpine Tethys rifting; 201-169 Ma) and the Late Jurassic-Early Cretaceous (Vocontian-Valaisan rifting; 150-129 Ma), followed by thermal relaxation and burial in the Alpine foreland during the Cenozoic (Two Rifts-model).

The model calculates for each tectonic scenario the evolution of the temperatures in a stratigraphic column according to the reconstructed burial history. The predicted maximum temperatures and are then compared to the $T_{max}$ measured at a given depth. The No Rift-model assumes a thermally-equilibrated post-Variscan European lithosphere characterized by crustal thickness of 30 km and a mantle lithosphere of 90 km (e.g. Mouthereau et al., 2021). For the second and third scenarios accounting for lithospheric thinning, we tested thicknesses variations ranging from 30 to 8 km, and from 90 to 15 km, for the European crust and mantle lithosphere respectively. As the margin is inverted during the Cenozoic, crustal and mantle thicknesses are designed to progressively increase to reach their initial pre-extensional values.

In all models, one of the major unknowns is the thickness of sediments (mainly Cenozoic), on top of the sections that have been removed by erosion. In the Barles Half Window (Fig. 1B), partially reset AFT ages documented in Miocene marine molasses reveal maximum burial depths of 3-4 km (Schwartz et al., 2017), an estimate that includes the thickness of the overlying nappe. Further to the SW, beneath the Digne Nappe, the thickness of the Mio-Pliocene series is constrained by the Mirabeau well to 2 km (Fig. 1B, see Graham et al., 2012). According to other thermochronological data obtained in Paleogene sediments in the vicinity of DEV, CLN and SLC sections and farther to the East (Labaume et al., 2008; Jourdan et al., 2013; Schwartz et al., 2017), the domain ranging between fully and partially reset AFT ages defines a region where sediments were heated between 90 and 120°C (Fig. 1B). We therefore infer that a maximum of 3 km of Cenozoic sediments have been eroded at the Digne thrust front (Fig. 1B). In addition, we have also evaluated the amount of missing Cretaceous strata in DEV, SLC and CLN sections based on the surrounding areas where they are preserved. Because all the sections are detached in the Upper Triassic evaporites, which were also involved in salt tectonics since the Early Jurassic (e.g. Célini et al., 2020), it is difficult to estimate their original thickness. We thus assume for every section a reasonable initial thickness of 100 m for the Upper Triassic evaporites. A thickness of 100 m of Lower Triassic sandstones and 50 m of Middle Triassic limestones have been considered based on the thickness exposed in a few location of the SW Alps, including the Lower and Middle Triassic exposed north of Barles in Verdaches (e.g. Haccard et al., 1989).

Physical and thermal properties of the rocks adopted in our models are presented in Table 1. These parameters have been chosen to reproduce the lithologies documented in the sedimentary cover (sandstone, limestone, marls, etc.) and assuming a continental basement with a homogeneous granitic composition. The temperature at the base of the conductive continental lithosphere is fixed at 1333°C (e.g. Mouthereau et al., 2021).

| Lithology | Density (kg.m⁻³) | Thermal conductivity (W.m⁻¹.°C⁻¹) | Heat capacity (J.kg⁻¹.°C⁻¹) | Radiogenic production (W.m⁻³) |
|---|---|---|---|---|
| Sandstone | 2670 | 4,6 | 740 | 9,5E-07 |
| Limestone | 2710 | 3,57 | 795 | 6,2E-07 |
| Marls/limestone alternation | 2690 | 2,76 | 805 | 9,6E-07 |
| Marls | 2680 | 2,76 | 815 | 1,3E-06 |

| Conglomerate | 2350 | 3,27 | 812 | 1,6E-06 |
| --- | --- | --- | --- | --- |
| Salt | 2160 | 6,1 | 865 | 1,0E-08 |
| Crust | 2650 | 3 | 1150 | 2,0E-06 |
| Lithospheric Mantle | 3350 | 3,3 | 1200 | None |

*Table 1: Rock properties adopted for modelling with TemisFlow.*

## RESULTS

### RSCM Temperatures

RSCM temperatures obtained along the six sections are presented in Fig. 3A (see also details in the Suppl. File 1). We systematically observe two temperature trends separated by an abrupt shift of c. 50 to 100°C (peak temperature drops in Fig. 3B). In the deeper domain, $T_{max}$ measured in Early-Middle Jurassic deposits at reconstructed burial depths between 2 and 4 km are ranging between 200 and 340°C, except for the CAS section (Fig. 3A). In contrast, upper successions of the Upper Jurassic and Cretaceous post-rift strata and the syn-orogenic deposits show $T_{max}$ below 150°C (Fig. 3). It is worth noting that the 100°C $T_{max}$ drop in section DEV occurs over a few tens of meters, corresponding to a sharp transition in the Middle to Upper Jurassic interval (Fig. 3A). Along the southern CAS section, located in the Castellane Arc, the $T_{max}$ do not exceed 170°C at the base of the section. Temperatures from Middle Jurassic series located farther to the east in the Embrun half window range between 300 and 330°C (Fig. 1B). Samples in the Jurassic and Eocene successions from the E-U nappes that are overthrusting the Digne Nappe (Fig. 1D) yield in contrast, notably lower $T_{max}$ between 190 and 225°C (Fig. 3C).

### Evidence for rift-related thermal event recorded at the Digne frontal thrust

The $T_{max}$ measured in the uppermost samples of the stratigraphic sections are in agreement with AFT ages from the Eocene strata (CAS, DGN) which show partially reset ages and thus temperatures below 120°C (Fig. 1B; Labaume et al., 2008; Jourdan et al., 2013; Schwartz et al., 2017). Assuming a post-extension thermal gradient of 30°C/km, in line with the upper bounds of published gradients in the SW Alps (e.g. Bigot-Cormier et al., 2006; Valla et al., 2011), $T_{max}$ of 100-150°C found in the upper parts of the studied sections are likely to reflect burial in the Alpine foreland. However, a similar burial with the same geothermal gradient fails to explain the high $T_{max}$ up to 300-330°C found in deeper Early-Middle Jurassic strata (DEV, SLC, CSN, DGN, CLN

sections) as they would imply 6-7 km of foreland sediments at the thrust front and the reset of all AFT ages. In addition, the abrupt ~100°C decrease of RSCM temperatures across the tectonic contact between the Digne Nappe and the E-U nappes (Fig. 3C) suggests that the emplacement of the E-U nappes of the Alpine accretionary wedge is not responsible for the high $T_{max}$ observed in the Digne Nappe. If it was the case, we would expect continuously decreasing temperatures across the contact. Rather, these higher temperatures in the Digne Nappe likely represent an older and higher-grade thermal event, which produced temperatures too high to be reset by the subsequent Alpine burial. A notable exception is the CAS section in which $T_{max}$ are mainly below 150°C and could therefore represent burial in the Alpine foreland only.

**Modelling results**

The comparison between the temperatures derived from RSCM analysis and temperatures predicted for the different scenarios based on TemisFlow® 1D modelling is presented in Fig. 4. To quantitatively evaluate our results, we calculated the root mean square (RMS) data misfit error between model and observations (Fig. 4). From burial histories modelling we obtained maximum temperatures and geothermal gradients in the stratigraphic columns, shown in Fig. 5 and Fig. 6 for the third model only (Two Rifts-model), which best fits the measured RSCM temperatures. Refer to the supplementary files for burial histories and geothermal gradients obtained for the two other scenarios (No Rift- and One Rift- models).

In the first No Rift-model, the $T_{max}$ are reached during syn-orogenic burial in the Alpine foreland. The fit is good for the uppermost samples but the model fails to reproduce the observed $T_{max}$ above 200°C measured in deeper samples (RMS ranging from 20 to 77°C) (Fig. 4 & Suppl. File 2). The only exception is the CAS section where the sample show $T_{max}$ distinctively colder (Fig. 3 and 4). For the second One Rift-model that incorporates crustal thinning to 8-10 km and shallowing of the LAB from 120 to 25 km during the Early-Middle Jurassic, the fit is noticeably better (RMS ranging from 17 to 53°C) (Fig. 4) for the deeper samples with higher temperatures (Fig. 4 and Suppl. File 3). This is caused by the increased of the crustal geothermal gradient from 70 to more than 90°C/km that results from lithosphere thinning (Suppl. File 4). The third Two Rifts-model (Fig. 5), which includes the superimposition of two rifting events in a few tens of million years, presents a much better fit of the temperatures both for the shallower and deeper samples with noticeably lower RMS in the 13-32°C range (Fig. 4).

Burial histories obtained from the Two Rifts-model show that the deepest sediments record $T_{max}$ between 230-330°C during the Early Cretaceous (~130 Ma, Fig. 5). As a result of the lithospheric thinning, crustal geothermal gradients increased from 50-60°C/km during the Early-Middle Jurassic to 80-90°C/km during the Early Cretaceous (Fig. 6). The cumulated thermal effect of the Jurassic and Early Cretaceous rifting

265   is required to fit the RSCM data, especially to reproduce the temperatures below 2 km (Fig. 4). The last rifting event is followed by thermal relaxation as geothermal gradients return to their initial value of 30°C/km due to deepening of the LAB 1333°C isotherm from the Late Cretaceous. Most of the reconstructed $T_{max}$ in the

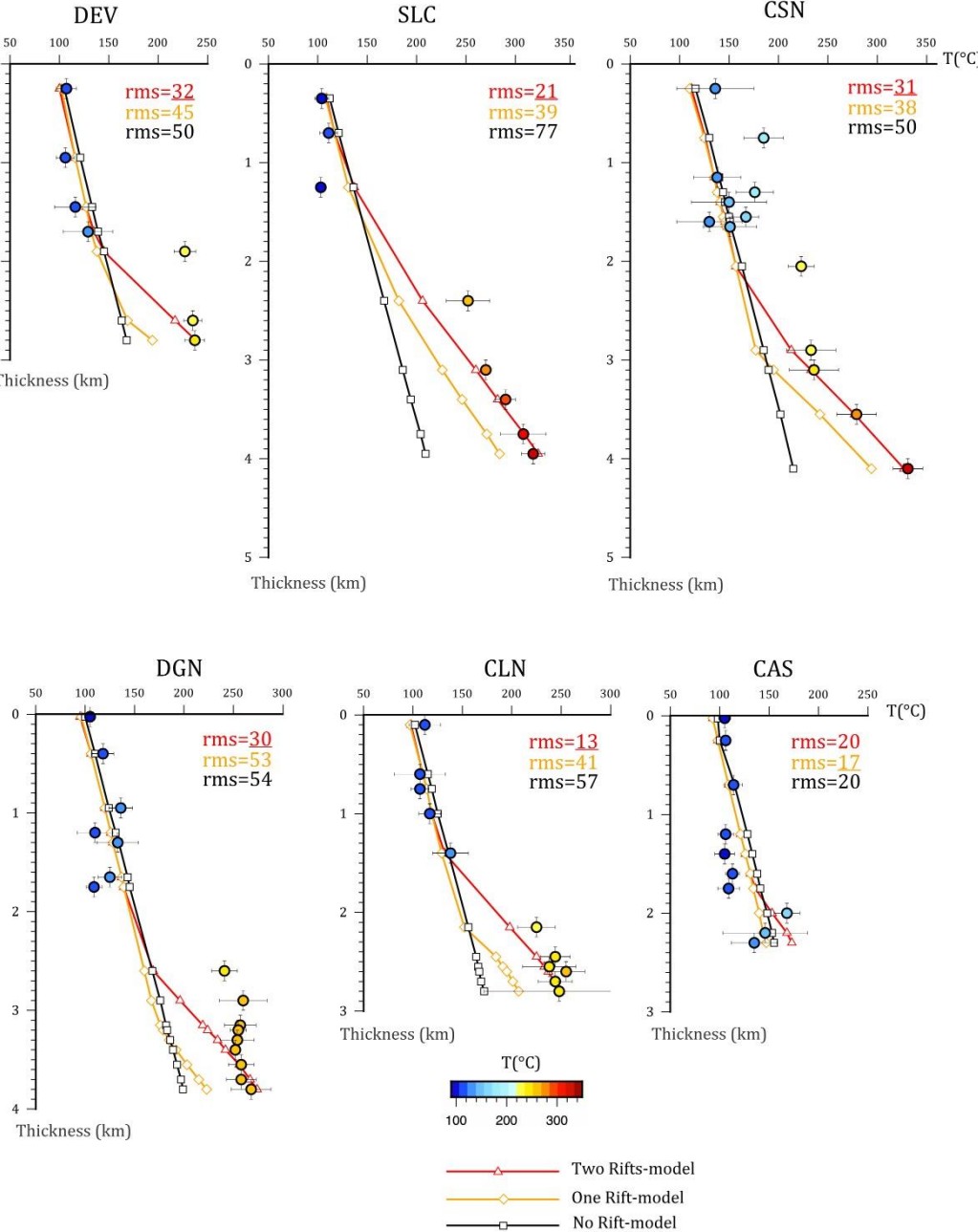

*Figure 4: Comparison between observed RSCM temperatures (colour-coded circles) and temperatures computed based on tested tectonic models (Two-rifts, One-rift and No-rift) represented by color-coded lines and symbols. RMS are the root mean square data misfit error calculated for each model. Reconstructed temperature-depth slopes correspond to thermal gradients: 30°C/km (upsection) and 80-90°C/km (downsection). RSCM data are best reproduced for the Two-rifts model and if peak temperatures are achieved during the Early Cretaceous.*

stratigraphic sections agree with the hypothesis of 3 km of eroded Cenozoic deposits. The only exception is the CSN section, which has required 4 km of burial in the foreland to fit the $T_{max}$ of the upper part of the section. This value agrees with a thermochronological study on Eocene-Oligocene deposits indicating that these syn-orogenic deposits can have reached around 4 km in that location of the Alpine foreland (Fig. 1B, Labaume et al., 2008). The temperatures of the colder CAS section can be accounted for by any of our scenarios with

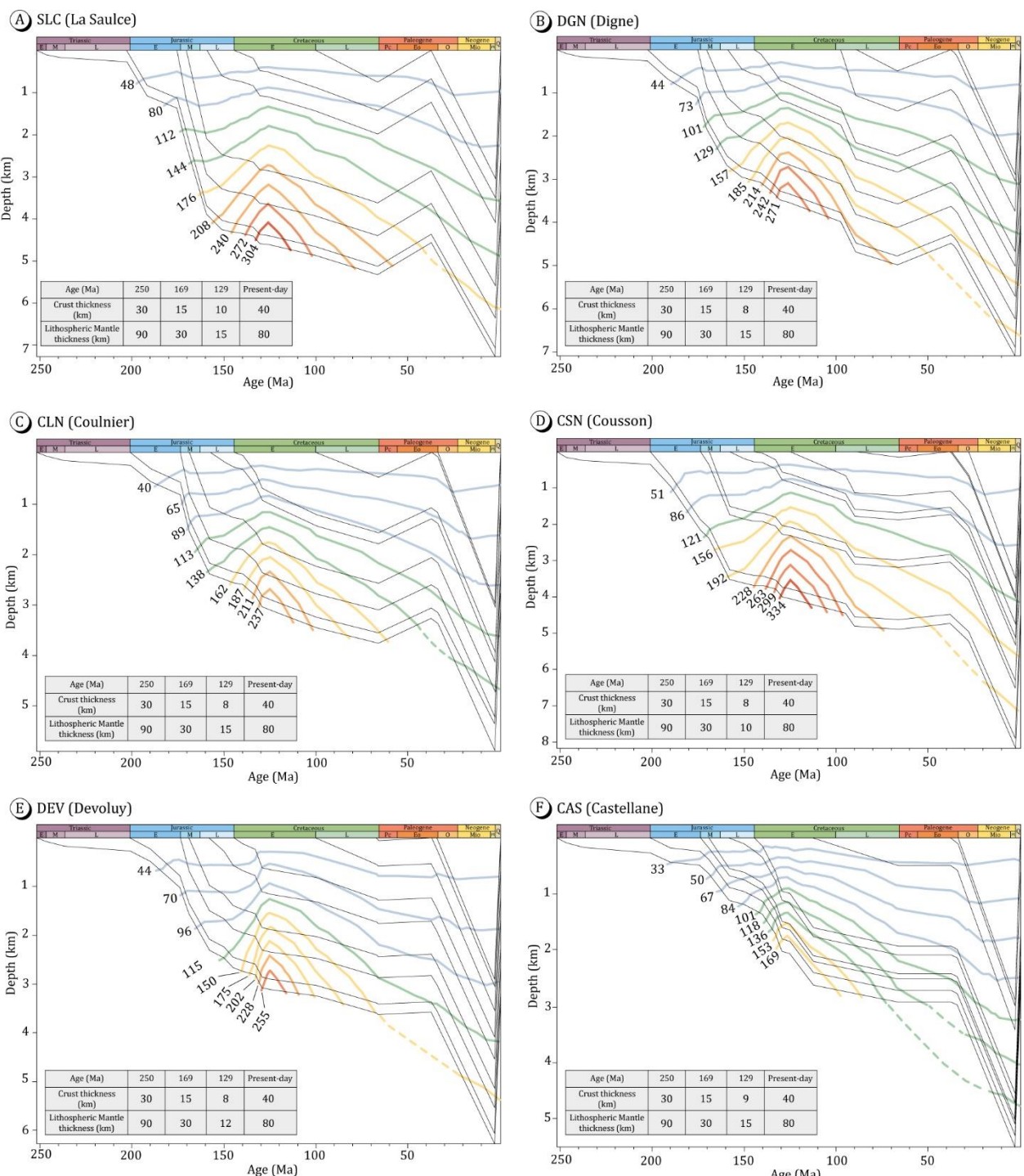

*Figure 5: Burial histories from decompacted stratigraphic columns generated with TemisFlow® for the Two Rifts-model. See location of stratigraphic sections in Fig. 1 and their detailed stratigraphy in Fig. 3.*

relatively low RMS values (Fig. 4). Although this section is noticeably thinner that the others, the burial history of CAS is characteristic of the Dauphinois palaeogeographic domain (Fig. 5F and Fig. 6F), as shown by the continuous subsidence from the Early Jurassic to the Early Cretaceous. We infer that the temperature evolution of the CAS section is consistent with the Two Rifts-model.

It should be noted that the evolution of geothermal gradients as a function of burial history locally reveals greater gradients for model cells located upwards (Fig. 6). This reflects the variability of thermal conductivities in sedimentary facies resulting in contrasting local thermal gradients. These numerical simulations also reveal that the higher thermal gradients computed in the upper part of the section are maintained for a longer period of time that is before gradients reduces to 30°C/km during the Cenozoic (Fig. 6).

## DISCUSSION

### $T_{max}$ explained by lithospheric thinning

In our preferred Two Rifts scenario, higher $T_{max}$ observed in the Digne Nappe are reached during the second phase of lithospheric thinning of the European margin during the Early Cretaceous, when the overburden is 3-4 km thick and geothermal gradients in the upper crust reached 80-90°C/km (Fig. 6). Only a limited combination of crustal thicknesses of 8-10 km and lithospheric mantle thicknesses of 10-15 km (Fig. 5) can reproduce the RSCM data. This scenario implies a significant thinning of the European lithosphere from 120 to 20-25 km resulting from a protracted period of stretching lasting from the Early-Middle Jurassic (201-169 Ma) to the Late Jurassic-Early Cretaceous (150-129 Ma). The total crustal stretching deduced from our models (β-factor) is 3-3.75 and reaches 6-9 for the lithospheric mantle, thus suggesting a depth-dependant stretching mechanism. We note that the crustal stretching factor of 3-3.75 fits well with the crustal thicknesses typically found at the transition between necking and hyper-thinned domains of magma-poor rifted margins (e.g. Tugend et al., 2015). However, our best Two Rifts-model does not require mantle exhumation.

Thermo-mechanical models of collisional orogens show that high gradients of 80°C/km inherited from the rift evolution are preferentially preserved in the less deformed retro-wedge side located on the upper plate (e.g. Jourdon et al., 2019; Ternois et al., 2021). Our study shows that this may also occur in the lower plate at the front of orogenic wedges where crustal thickening is not sufficient to overprint the rift-related thermal peak.

**Tectonic reconstruction of Early Cretaceous rift in the SW Alps**

Tectonic and palaeogeographic reconstructions of the SW Alps (Fig. 7A) reveals that Jurassic – Early Cretaceous successions of the Digne Nappe were originally located to the East of the Vocontian Domain (Beltrando et al., 2012; Dumont et al., 2012; Angrand and Mouthereau, 2021; Lemoine et al., 1986), in the continuation of the Valaisan Domain and separated from it by a NW-SE transfer zone (Lemoine et al., 1989).

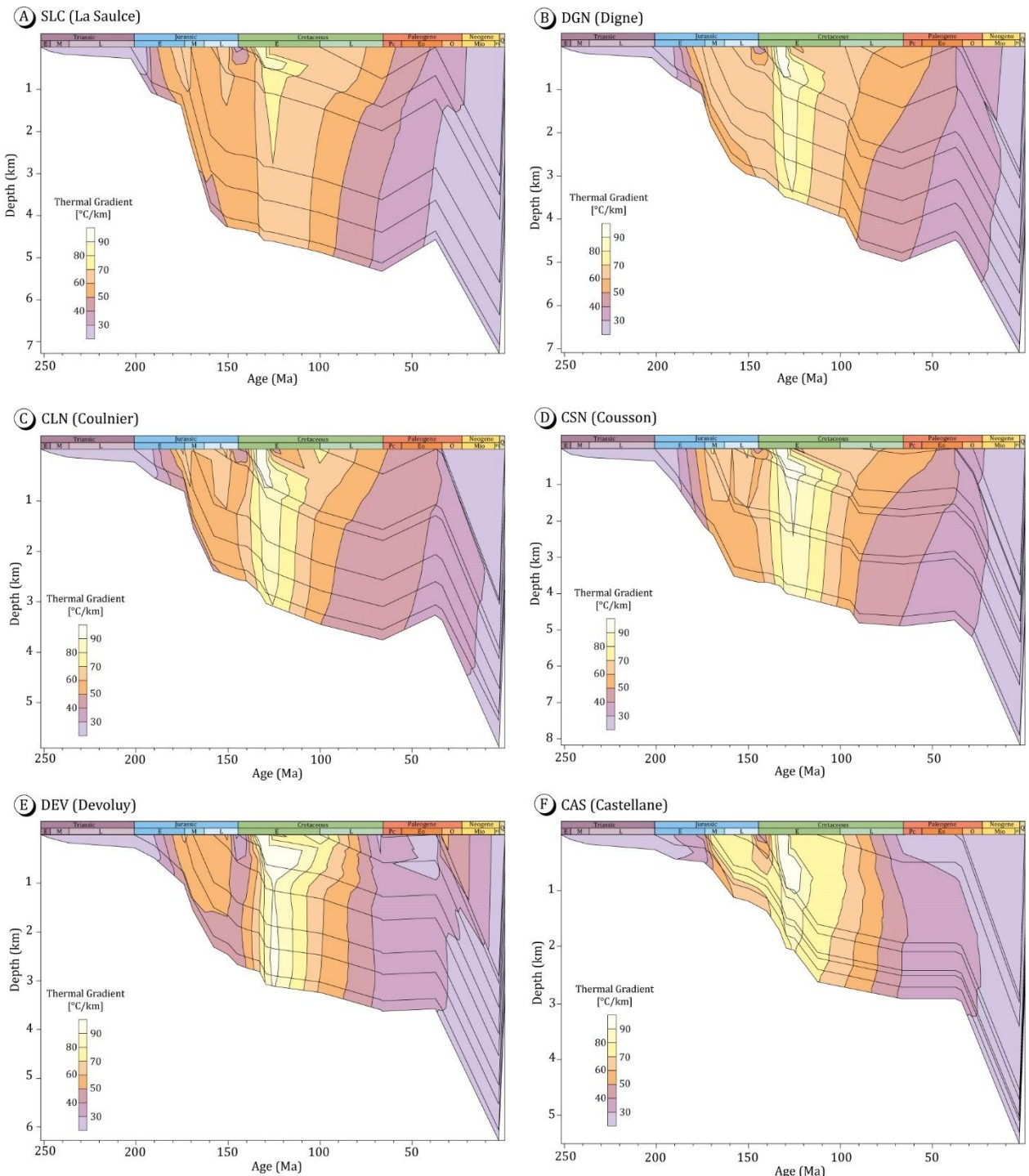

*Figure 6: Burial evolution of each sedimentary column and evolution of geothermal gradients along the sections for the Two Rifts-model.*

The existence of this transfer zone is justified by the large extension accommodated in the Valaisan Domain as indicated by mantle exhumation and magmatism, that strongly contrasts with the lower extension in the area of the Vocontian Domain and the Digne Nappe where a remaining crustal thickness of 10km is proposed (Fig. 5 and Fig. 7B). The transfer is aligned in the direction of extension along the eastern borders of the Argentera and Pelvoux massifs (Lemoine et al., 1989). The lack of continuity of the Valaisan Domain towards the SW also points to this.

Numerical models of propagating rifts and spreading ridges show that such transitional domain can localize extension for tens of millions years before the rift actually propagates into the continent (Jourdon et al., 2020). The different duration of rifting between Vocontian and Valaisan domains provides a good analogue of such 3D architecture of extensional systems. The end of rifting in the Vocontian Domain/Digne Nappe area is indeed constrained in our model to be around 130 Ma, but extension in the Valaisan Domain is suggested to have persisted until 110-90 Ma. Rift propagators are also sites of asthenospheric flow oriented parallel to the rift axis (Mondy et al., 2018). In this case of westward propagating rift in the SW Alps, the asthenospheric mantle flow and lithosphere thinning are likely to transfer heat laterally from the Valaisan to the Vocontian domains. Accounting for these three-dimensional thermal effects should potentially reduce the lithospheric stretching needed to fit the data, and this is yet to be evaluated.

In the reconstruction of Fig. 7 the flyschs deposits of the E-U nappes are supposed to be originally deposited over the Alpine Tethys, the Briançonnais and the Valaisan domains, which corresponds to the current position of the internal zones. The tectonic thickening of the E-U nappes reaches a maximum in the East of the Embrun half window (Fig. 1B), where $T_{max}$ measured in the E-U nappes and the Digne Nappe are very close (Fig. 3C). These temperatures thus can be explained by a nappe stacking of ca. 10 km when approaching the internal zones under a normal gradient of 30°C/km. The $T_{max}$ difference observed at the NW front of the E-U nappes (Fig. 1B and 3C) however reflects the overthrusting of the internally-derived E-U nappes and also requires heating of the Jurassic successions before the emplacement of the E-U nappes. Based on our analysis of the Digne main thrust, those mid-crustal temperatures may reflect pre-Alpine heating, and temperatures inherited from the same Early Cretaceous thermal event. This implies that the thermal anomaly would not only be restricted to the Digne thrust front but must be of regional importance extending farther to the East (Fig. 7).

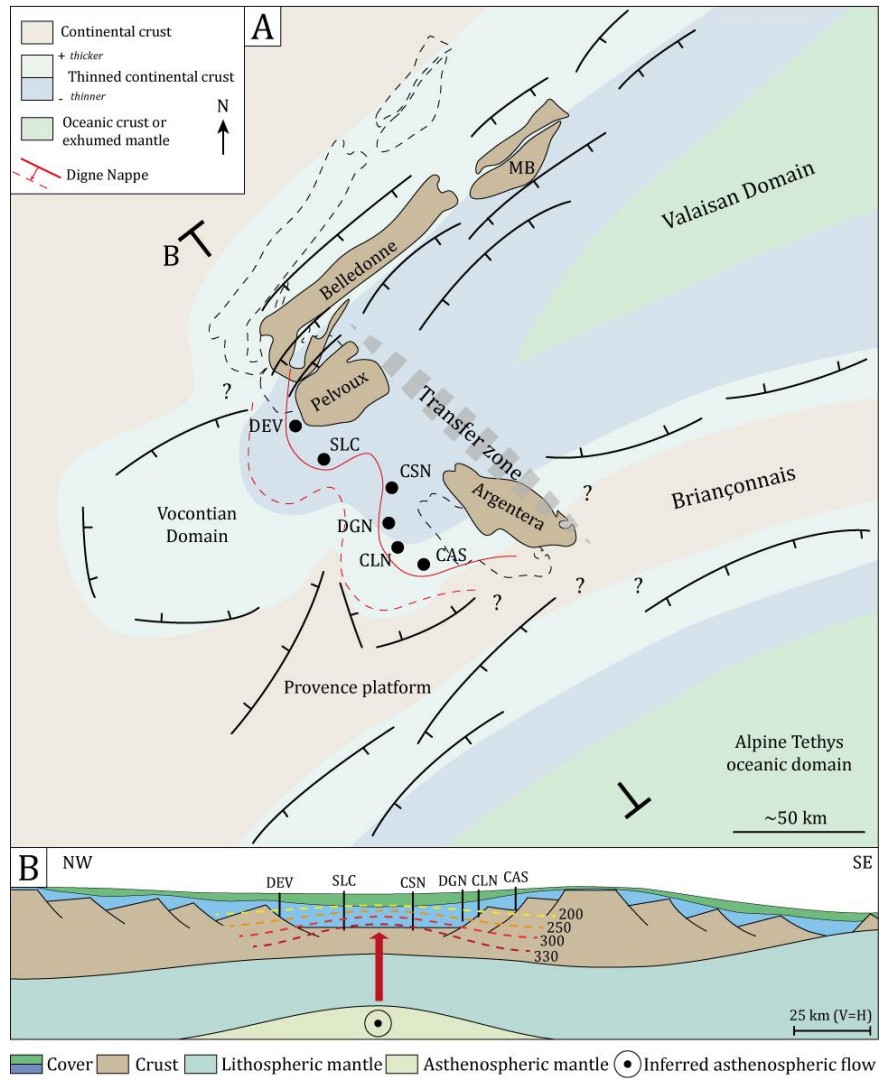

*Figure 7: (A) Palaeogeographic reconstruction of the European margin in the Western Alps, during the Early Cretaceous after Beltrando et al., 2012; Dumont et al., 2012; Angrand and Mouthereau, 2021; Lemoine et al., 1986. Restored positions of the external crystalline massifs are after Bellahsen et al. (2014) and the Digne Nappe is restored based on shortening estimates from Ford et al. (2006). The SW Alps and the Digne Nappe were located in the continuation of the Valaisan Domain, as a result of the W-directed propagation of the Early Cretaceous rift system into the Vocontian Domain. Dashed lines represent the present-day positions of the External Crystalline Massifs and the Digne thrust front. MB = Mont Blanc massif. (B) Schematic reconstruction of the rifted margin through the study area, see location in (A). Dashed lines are notional isotherms (°C).*

### Rift-related metamorphism in the SW Alps?

The significant thinning of the lithospheric mantle is the dominant factor explaining the thermal (Fig. 5) and geothermal (Fig. 6) structures reached during the Early Cretaceous. The high geothermal gradients of 80-90 °C/km inferred from the modelling (Fig. 6) have been recognized in other inverted rift basins of Europe. This is the case for the Early Cretaceous Pyrenean rift system (Vacherat et al., 2014; Hart et al., 2017; Saspiturry et al., 2020; Clerc and Lagabrielle, 2014; Clerc et al., 2015) where such geothermal gradients are accompanied by high-temperature metamorphism, hyper-extension and mantle exhumation (e.g. Ducoux et al., 2021). Laterally, to the West, it leads to continental break-up and oceanic spreading in the Bay of Biscay. Similarly, high

geothermal gradients have also been reconstructed in the Cameros basin in the Iberian Range (Rat et al., 2019) where metamorphism and hydrothermalism have been recognised.

Temperatures locally above 300°C found at the front of the Digne Nappe (Fig. 1D and Fig. 3) are consistent with previous estimations of illite crystallinity index (Aprahamian, 1988) and the proportion of chlorites in clay mineral assemblages recognised in the Mesozoic formations of the region (e.g. Artru, 1967; Levert, 1991). They are indicative of diagenesis to weak metamorphism (anchizone). The pressure-temperature conditions for regional greenschist metamorphism are not met in the Digne Nappe at the time of Tmax

acquisition due to low lithostatic pressures (0.9-1.2 kbar) and the lack of metamorphic fluids at the base of the nappe.  Mineralisation of barite, authigenic quartz and pyrite in the Callovian-Oxfordian shales of the Terres Noires formation (Fig. 1C) are described locally in both the hangingwall and the footwall of the Digne thrust (Guilhaumou et al., 1996). They form mineral concretions from which fluid inclusion in quartz revealed temperatures up to 250°C and pressures of 0.6-0.8 kbar. These local fluids are interpreted as reflecting the

maximum burial of the Jurassic black shales acquired during the Middle Cretaceous (Guilhaumou et al., 1996), thus confirming our results. We infer that the groups of homogeneous temperatures found at the base of the sections might have required localised influx of hot fluids to allow an efficient heat transport in the sedimentary cover but did not result in rift-related metamorphism.

**CONCLUSION**

    The RSCM approach has been applied to 90 organic-rich samples collected along the front of the Digne Nappe, in its autochthonous and in the Embrunais-Ubaye nappes. Paleotemperatures of about 300°C reported at the front of the SW Alps are close to or even higher than paleotemperatures reported from the internal domains (e.g. Briançonnais). Basin modelling has been used to test three scenarios (No Rift-, One Rift- and Two Rifts-

models) to reproduce the Tmax. The results of numerical modelling reveal that the Two Rifts-model provides the best fit for the observed temperatures. These mid-crustal temperatures are interpreted to be inherited from a long-lasting lithospheric stretching process between Europe and Adria/Iberia. $T_{max}$ were reached during the Early Cretaceous (~130 Ma) corresponding to geothermal gradients of 80-90°C/km, that are not related to crustal thickening during orogeny. Our study argues that the recognition of collisional thermal events in accretionary

prisms may be elusive without a careful analysis of the pre-orogenic thermal evolution of the accreted units. $T_{max}$ reconstructed in orogens may therefore not illustrate nappe stacking or maximum syn-orogenic burial in

the foreland, but instead could represent pre-orogenic thermal conditions. This can lead to overestimating both orogenic thickening and horizontal displacement in mountain belts. This issue is likely to be more significant in older orogenic systems where details of the original structural relationships are lacking. In the case of the Alps,

the identification of an Early Cretaceous thermal event introduces new constraints on the palaeogeographic and kinematic reconstruction of the Alpine Tethys, and on the rift architecture.

## DATA AVAILABILITY

All the data used in this manuscript are available in the supplementary files and can be provided by the
corresponding authors upon request.

## AUTHORS CONTRIBUTION

Naïm Célini: Investigation / Conceptualization / Visualization / Writing Original Draft / Formal Analysis.

Frédéric Mouthereau: Investigation / Conceptualization / Funding Acquisition / Writing Review and Editing.

Abdeltif Lahfid: Investigation / Formal Analysis / Resources / Funding Acquisition.

Claude Gout: Software / Resources / Writing Review and Editing.

Jean-Paul Callot: Investigation / Funding Acquisition / Writing Review and Editing.

## COMPETING INTERESTS

The authors declare having no competing interests.

## FINANCIAL SUPPORT

The work has been supported by the Agence Nationale de la Recherche, Carnot Institute ISIFOR and BRGM.

**ACKOWLEDGEMENTS**

We wish to thank Yann Rolland and two anonymous referees for their thorough comments. Mary Ford is also thanked for her comments on an earlier version of the manuscript. Beicip-Franlab is acknowledged for providing licences of the basin simulator TemisFlow™.

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
