# Peer review of "Rift thermal inheritance in the SW Alps (France): insights from"

_EGUsphere, 2022_

## Referee Comment (RC2)

[referee-annotated manuscript omitted]

---

## Author Comment (AC1)

Answer to the comment https://doi.org/10.5194/egusphere-2022-949-RC1, 2022

First, the co-authors would like to thank the Anonymous Referee for its thorough comments on the manuscript.

All the modifications suggested regarding the typo have been done directly in the text.

I read this work with great interest. The starting hypothesis of recognizing a signal of pre-collisional thermal events through the use of geothermometers in an external sector of the chain (where therefore the Alpine metamorphism may not have overprinted everything) seems interesting and promising, possibly providing important contribution to the reconstruction of the pre-Alpine paleogeography evolution of this sector of the Paleo-European continental margin.

However, I believe that a lot of work still needs to be done in order for it to be published. Some important points need to be addressed by the authors; among these, the most important concern the controversy on the nature and timing of the Valais oceanic domain (ignored by the authors) and the fact that the high temperatures presented in this work should imply, in my opinion, some evidence of metamorphism, in a domain which is, on the contrary, commonly described as non-metamorphic (in my extended comments below you will find much more comments on these points).

Furthermore, in many parts the data presentation can be improved, making it more organized and understandable. In the Interpretation and Discussion sections some sentences are a little obscure and hard to understand, and some points deserve wider explanations and discussions. I got the impression that it was written a bit in haste, without the necessary accuracy and without providing a deep and complete discussion of the reliability of the measured data and of their geological meaning in the Alpine geological context.

For these reasons I think that the manuscript in its actual version can not be published, and recommend major revisions.

**Abstract**
Lines 14 to 17: you should better specify where the samples come from. The expressions "along the Digne nappe" or " the nappe stack" are too generic
L 14 and 17: We have added complementary informations on sample location.

**Introduction**

Line 30: no comma needed after "crustal structure"
L 30: Done.

Line 49: the sentence "…together with issues realted to the magnitude of continental collision" is too generic, you must specify and/ore quote some papers

Line 51: Why "Although"? The second part of the sentence is not in contrast with the first…
Line 49 and 51 : Corrected. We have reorganized the introduction and reformulate some of the sentences.

Line 54: "carrying": better "Characterized by"; why "..a thick Jurassic syn-rift section"? The Digne nappe is not made up of uniquely Jurassic successions....
Line 54: Modified.

Line 56: what do you mean with "pre-collisional stratigraphic series"? The successions (succession is more appropriate than serie) underlying the Alpine Foreland one? But in some cases you sampled up to the Eocene.....
Line 56: We have replaced "series" by "successions" and modified the text to clarify the position of the studied stratigraphic units relatively to the nappe emplacement and other main stages of the tectonic evolution of the margin.

**Geological Setting**

The authors write about two rifting events, stating that in the Late Jurassic – Early Cretaceous the Valais ocean opened. They describe this in few words, as it was a fact, well documented and acknowledged by all the geological community. But it really is not. There has been a long controversy on the Valaisan, and still there is, that concerns its nature of oceanic basin, its extension, and its timing. The Authors quote the paper by Beltrando et al., 2012, but they do it inappropriately. Beltrando et al. dedicated a lot of space to reviewing the debate concerning the Valais (not by chance the title of the paper start with "The Valaisan controversy revisited..."), and in the discussion they interpreted they original data suggesting that "the crustal thinning in the Valaisan basin may have been kinematically linked with the opening of the Western Tethys" (thus not in the Cretaceous!). Geochronological data are scarse and often contrasting (e.g.: Liati et al. 2005), and many Autors provided a large body of structural and stratigraphic evidence pointing to a Jurassic opening of the Valaisan basin, even in recent papers (see for example Mohn et al. 2010, Ribes et al., 2019, 2020). This is not my research field, and I can not say if the Valais opened in the Jurassic together with the Western Tethys or later in the Cretaceous. But the authors can not ignore that it is still a highly controversial topic, and should report and comment on this. Actually, if the authors introduced it adequately, the data resulting from this paper could give a new contribution to such problem!

We agree with the reviewer. We have reworked this part. The formation of the Valaisan domain, based on existing geochronological and field constraints can broadly agree with the timing of the formation of the Western Tethys. But it also recorded subsidence, magmatism/metamorphism during the early Cretaceous. We therefore emphasize that the European margin is the result of a long lasting evolution since the early Jurassic marked in the Valaisan Domain by an early Cretaceous thermal/metamorphic and magmatic event.

Lines 63 to 78: the first sentence is not clear, should be rephrased. In general, the description is not always linear (see for example line 68, with an abrupt jump from the second rift event to the continental breakup, which however is referable to the first one). And the 33.9 Ma age seems to me too precise.... Do the quoted papers really indicate it for a long-lasting event such as the transition between two collisional stages?
L 63 to 78: We reorganised and corrected the text.

Line 81: "hangingwall" is unnecessary
L 81: Modification done.

Line 84: "It turns..." refers to the main thrust. But in the previous sentence you were referring to the Digne nappe.... Please rephrase.
L84: rephrased.

Line 85: A nappe is not a thrust front....
L 85: We keep nappe and corrected the rest.

ines 85 to 88: please use succession instead of serie. A description of the succession should be provided. It can be brief, but I think it is necessary, since you collected your samples all over it.
L 85 to 88: modifications done through the text. We added a brief description of the dominant lithologies.

**Samples and Methods**
**RSCM Thermometry along reconstructed vertical sections**

This section is definitely incomplete and bad organized. As general comments I would say that authors should first of all briefly explain what RSCM thermometry is and how it works, since this journal is not uniquely dedicated to Raman analysts (and please, the first time you cite RSCM wrote it in the extended version). They should then describe which was the goal of the sampling and analysis, and provide a lithological (and possibly a petrographic one too) description of the samples (they wrote carbonate deposits: too generic, I guess thay should be CM-rich rocks.... Are all limestones or shales too?). And some description of the analytic instruments and their operating conditions must be provided. Lines 105 to 112 do not refer to the methods of this work and should be moved to the Results or Discussions sections.

The first time we cite RSCM is in the Abstract and we wrote it in its extended version. We thus did not modify this in this part of the text.

We added a paragraph stating what RSCM thermometry is and which material has been used.

We also moved the paragraph from L 105 to 112 in the Results section.

Line 97: the first time please write the name of the sections in the extended version. The same in the legend of figure 1B
L 97: done.

Lines 100 to 102: This sentence ("Because the nappe.... prior to thrusting") is definitely not clear, please rephrase
L 100 to 102: Agreed. We modified the sentence.

Lines 105 to 109: too long sentence, better to split in 2 or 3. The statement "during the passive margin stage coeval to crustal thinning" is wrong: the passive margin stage goes on well after the end of the crustal thinning.
L 105 to 109: modified for more clarity.

Lines 109 to 112: This is an important point for interpretation of your data and I think deserve more explanations. Anyway, not in this section (see comment before)
Agreed.

**Numerical modelling with basin model**

Line 116: "temperatures across the Digne thrust front.." did you mean along?
L 116: yes we did, we modified.

Line 117:  convert in place of converted

L 117: We modified the sentence.

Line 118: I do not understand "burial in the Alpine foreland"... In this scenario the Tmax is achieved due to the unique effect of the burial history. But during the entire burial history...
L 118: Yes we clarified this in the text.

Line 120 and line 124: some values of the crust and lithosperic mantle are reported. How did you chose them? At least some citations are needed.
L 120 and 124: This is now better presented in the revised version. We have chosen initial boundary conditions, prior to lithospheric thinning, as follows, 30 km for the crust and 90 km for the mantle lithosphere. These values are typical for the post-Variscan lithosphere (e.g. Mouthereau et al., 2021).

Lines 125 to 130. All this explanation is not completely clear, particularly the last sentence. How could you reconstruct the thickness of the Cretaseous sediments eroded during the formation of the Foreland basin unconformity starting from the AFT data in the Miocene deposits? Please try to better illustrate your reasoning
L 125 to 130: We rephrase this part. We do not use AFT ages in the Miocene to reconstruct the Cretaceous. We have estimated for DEV, SLC, and CLN sections the amount of eroded Cretaceous (coherent with what is observed in surrounding areas where it is not eroded) and the 3 km of Cenozoic which is eroded as well.

Line 141: Why did you chose to fix the T at 1333 °C? And which is the reason for a such specific value? It is an assumption, so I do not understand which is the meaning of a such specific value. I mean, 1350 would be uncorrect? Why?
L 141: This value corresponds to the temperature below which heat is lost by conduction which is also the definition of the lithosphere. We rephrased this sentence.

Line 149. In which locations of the SW Alps is reported this thickness of Middle Triassic limestones? In many places (e.g. External Briançonnais) the Middle Triassic is much more thick....
L 149: This is the thickness of the Middle Triassic reported in the Verdaches area just to the North of the Barles half window, at the base of the Digne Nappe which is the study area. This value is reported from the "La Javie" geological map at 1/50000 (Haccard et al., 1989) which mentions a multi-decametric thickness of Middle Triassic limestones in Verdaches.

**Results**
**RSCM Temperatures**

Line 155: "Domain"?? I would replace with trend
L 155: We already used the term "trend" for the temperatures. We use the term "domain" when we talk about locations along the "vertical" sections.

**Evidence for rift-related thermal event**

Some major points corcernings this section.

It contains a discussion of the RSCM results, and thus in my opinion it should be moved to the Discussion section.

You found very high Temperatures, up to 340°, in the Early-Middle Jurassic beds of the most of the sections. These temperatures are commonly considered to be in the range of metamorphism. Low grade metamorphism (green schist facies), but metamorphism. In the Discussion below you propose an interpretation in which such thermal perturbation is related to two rifting events: this means that such perturbation lasted for tens of million of years. I would expect that sedimentary rocks affected by temperatures above 300°C for such a long time  interval in an extensional setting (in which the fluid circulation through the crust is highly favoured) would be transformed in metamorphic rocks. At least they should bear evidence of recrystallization and neo-blastesis. And I guess that many of your samples are shales or rocks with a shaly component, that is the most reactive to metamorphic reactions. You should face this point by providing a petrographic description of your samples. In the case you can not find any evidence of recrystallization and neo-blastesis you should propose a mechanism that hampered these processes at such high temperatures. I am not an analyst or a Raman expert and I do not want to doubt about the RSCM method; but I know that there has been (and probably there is) debate around it. Some authors reported kinetic effects and the occurrence of metastable poorly crystallized graphitic carbon that, in their opinion, would affect the reliability of the RSCM geothermometer (see for example Foustoukos 2012, American Mineralogist). All the more reason you should describe the rocks you analysed, showing the differences between the ones in the upper part of the sections which not experienced high temperatures and the ones in the lower part which were affected by temperatures up to 340°C. Alternatively, if you could not find any evidence, you should discuss how it was possible, in order to exclude that such temperatures are "fake" temperatures due to analytical artefacts.

Line 168. "(para-) autochtonous" I would avoid this old and ambiguous terms: you could more simply use "AFT ages from the Eocene sediments of the Digne nappe"
L 168: we modified as suggested.

Lines 173 to 175: I got your reasoning but I think you could make it a bit more explicit
L 173 to 175: We have elaborated on this and added a sentence to better explain our reasoning.

Line 174: Figure 2C, not 2B
Corrected.

Lines 177-178: see comment before too. Why do you talk about syn-orogenic burial? I would say that T values in the CAS section are consistent with a normal and continuous burial history (fromt Triassic to Pliocene)...
L 177-178: Yes, but please note that in this section we present the main characteristics of the RSCM temperatures before TemisFlow modelling. For CAS, temperatures are below 150°C. Given the thickness of the foreland basin sediments and a gradient of 30°C these temperatures could simply reflect burial in the foreland.  We have modified the text to make it clearer.

**Results of numerical modelling**

Line 185: did you mean figures 4 and 5? In figure 3 results from all the three models are shown
L 185: We start with Figure 3 which present a comparison of predicted and observed temperature of all the models (3 scenarios). Figures 4 and 5 correspond to the results obtained for the Two Rifts  model only.

Lines 195-196: "deepest sediments": which sediments are you referring to? The ones below 2 Km? In this case I would say between 230-340°C, not above
L 195-196: modified as suggested.

Lines 205 - 209: This is not convincing at all! The CAS section is just 24 km from the CLN section, and owns to the same tectonic unit and the same paleogeographic domain. Moreover, you can not desume a more internal location from the described difference in thickness of the Jurassic succession (and in figure 6 you do not locate it a more inner position....). Continental rifted margins are characterized by significant and abrupt changse in thickness of the syn-rift sediments. In the Dauphinois domain see for example the Ornon fault area (Chevalier et al 2003, and their figure 1c); the Briançonnais domain is characterized by a reduced and condensed Jurassic succession, but it pertained to a much more external position with respect to the study area of this work. I think that another explanation must be given for the "colder" values of the CAS section.

L 205 – 209: Note that in the study area the relationships between the syn-rift strata and its basement are not documented because the Upper Triassic evaporites have been decoupled during extension and involved in salt tectonics since the Early Jurassic. The facies and the thickness of the syn-rift strata in the Castellane are completely different from the ones in the northernmost area of the Digne Nappe which was located in the deeper part of the Digne-Gap basin during the Mesozoic (see Baudrimont & Dubois, 1977). However, it is true that the burial history looks very similar we therefore chose to adopt the same Two Rifts interpretation.

Lines 210 - 215: This is an important point and deserves much more attention. What are the 3D effects? And the kinetics of organic matter maturation? You simply mention these aspects, but I think you should comment them. Some authors do not consider the RSCM geothermometer reliable because of some of these problems (see my comment above): you should comment more extensively.

L 210 – 215: Thanks for this comment which lead us to look more carefully at existing data in the SW Alps. We are aware of the current debates on the importance of pressure and deformation on the reliability of RSCM geothermometer. The kinetics of organic matter maturation should also be taken into account when we interpret those data. We think that these temperatures are broadly in agreement with other previous publications in the region, specifically within the Jurassic "Terres Noires" which outline the significant role of fluid circulation during the Cretaceous. We have tried to make those points clearer in the new version. We think that the robustness of the rscm data is not in question and that it can be explained by other mechanisms such as the presence of fluids.

**Discussion**
**Tmax explained by decoupled crust-mantle**

Lines 219 to 226: such a huge lithospheric stretching and very high geothermal gradients should have a large body of evidence, even in the syn-rift stratigraphic successions (paleofaults, hydrothermal products, abrupt thickness changes, large breccia bodies, etc....). Actually some of these evidence have been reported in literature for the Early cretaceous too, both in the Dauphinois Domain and in the adjoining Brianònnais Domain. I think you should consider this.

L 219 – 226: We have provided more details from the literature in the discussion.

Lines 229 to 231: It is not clear which is the old proposed position of the Digne nappe and your new interpretation. Do you think it corresponded to a hyper-thinned domains? And why it should have been located on a transfer zone?

L 229 – 231: We removed this sentence. The existence of the transfer zone is now well accepted in communnity as indicated by the recent papers (Ribes et al., 2019;2020 ; Dall'Asta et al., 2022) and was actually first mentioned by Lemoine et al. (1989). The justification is first based on the difference of extension between the Vocontian and Valaisan domains, the fact it is parallel to extension direction and that the Valaisan is not exposed southwards.

Line 235: "expected to be better preserved": what should be better preserved? And the European paleomargin represented the lower plate during the collision....
L 235: We rephrased this part of the discussion.

Lines 236 to 240: Ok, but which could be the reason? Please provide an interpretation
L 236 to 240: Yes this is due the high conductivity upwards. We have reformulated.

**Tectonic reconstruction of Early Cretaceous rift in the SW Alps**

Line 250: what do you mean with "differential extension"?
L 250: we removed "differential".

Line 251-251: What the expression "In case of westward propagation…" mean? Do you think a westward propagation occurred or not?
L 251: Yes, we modified for more clarity.

Line 254: "To the east of the Embrun…": better in the eastern part of the Embrun…
L 254: Modified as suggested.

Line 257: why "in contrast"? I would write: "…reflects the overthrusting of the E-U nappes bu also require heating…"
L 257: modified as suggested.

General comment: some problems and doubts remain concerning the reliability and accuracy of the applied RSCM geothermometer (e.g.. the drop not reproduced by models; the "colder" CAS section; very high temperatures in a non-metamorphic domain). I would thus be more cautious in proposing such extremely high geothermal gradients, and at least I would suggest that it would be necessary and interesting to apply other geothermometers (illite crystallinity index?) in order to confirm or not the presented T values.
We have developed this issue in the discussion. We do not believe that the RSCM data should be questioned because solutions exist to explain this gap although they have not been tested in our approach presented in this paper.

Line 272: "foreland deposition"? not clear, please expand and clarify
Done

Figure 1

- A) It lacks a legend. And in the external zone you should more clearly distinguish the Digne nappe

- B) The geological map of the external zone is hardly readable. It is too detailed, with a lot of not useful (and not homogenously distributed) elements that, given the figure size, make the figure itself hardly readable. In the caption please add that 15 original RSCM analysis are indicated, and report the name of the sampled stratigraphic sections. Delete the second "reconstructed".

- In the legend the symbols for the tectonic contacts are missing. And I would highlight the Digne main thrust with initials

Modified.

Figure 2

- The font size seems to me too small. Particularly the numbers in 2A and the text in 2C, almost unreadable. In 2B it is not specified that the values of the columns are temperatures

The figure has been modified to improve its readability.

Figure 3

This figure is too small, and the text and numbers. Almost unreadable. You could split the 6 diagrams in 3 + 3....

The figure has been modified as suggested.

Figure 4

The isotherms lines are too thin, difficult to see them

Modified as suggested.

Figure 5
Specify in the caption that the models are based on the two-rift hypothesis

Modified as suggested.

---

## Author Comment (AC2)

**Answer to RC2**: ['Comment on egusphere-2022-949'](), Yann Rolland, 23 Oct 2022

**General comment**

I have read this manuscript with interest as it provides a synthesis of recently acquired (and some new) RSCM data at the SW Alps scale and provide a comprehensive modelling approach to explain these data. The form of the manuscript is excellent. It is well organized and concisely written, the figures are well drafted. Though, I am not an expert of modelling, I found this part well presented and understandable at least from the results.

I found especially interesting the position of the authors in not considering raman temperatures as just a result of Alpine thickening as previous authors did (see specific discussion points below), but as a result of previous rifting history. Of course, this could be surprising at first glance in metasediments not exhibiting any other metamorphic minerals at temperatures >300°C. However, this is always the case in Low Pressure sediments (like in contact-metamorphism in general), and maybe the impact of (hot) fluid circulations could explain some relatively short-lived thermal reset (note that ore formation is documented at least for the Jurassic phase). This point could be better discussed in the paper. However, I found that the explanations proposed (two rifting episodes) and tested by modelling are very interesting and fit very well with the regional geology where Jurassic and Lower Cretaceous rifting phases are clearly highlighted. Finally, the general model that is presented seems sound. Of course, the relationship with the Valais ocean is not simple, as the paleogeography has been significantly disturbed by the Alpine orogeny, but, however, the proposed reconstruction agrees with the shape of the Vocontian low and the dissymmetry of the Valais unit (not continuing in SW Alps).

In conclusion, I think that this work deserves a rapid publication with some relatively minor revisions.

Minor comments

Figures: in many cases the text in figures is too small (see attached file).

Figures have been checked and modified where relevant.

L47 : Altough, these temperatures are similar to those estimated by mineral thermometry in the Briançonnais (e.g. Lanari et al., 2014), this is no the case in the Pelvoux domain where RSCM estimates lie systematically higher, by about 50°C in eastern Pelvoux, (Simon-Labric et al., 2009), and by > 100°C to the west of Pelvoux ECM (especially, considering vitrinite reflectance results from Cretaceous beds,  ( Deville and Sassi (2006) ), which led Bellanger et al. to propose a "hypothetical" substractive contact that has not been observed in the field...

Agreed. We added some of the references suggested and tried to make clearer in the text the fact that temperatures in the external domains can be larger than those estimated in the internal zones.

L75 :

- I would say 'before 34 Ma' to be more consistent with error bars of geochronological methods (Ar-ar dating of syn-kinematic phengite)

We corrected this part as also suggested by RC2

- Top NW deformation is documented in the Eocene: Lanari et al., 2014, Terra Nova: Top NNW syn-kinematic deformation dated at 45 Ma in briançonnais units at the base of the Helminthoid flysch

We modified the text and added this ref.

L271: not sure the concept of "temperature structure" is adequate...? maybe the thermal state/profile?

We modified as suggested.

See attached file for details

---

## Author Comment (AC3)

Answer to the comment https://doi.org/10.5194/egusphere-2022-949-RC1, 2022

First, the co-authors would like to thank the Anonymous Referee for its thorough comments on the manuscript and its detailed reading of it.

**Abstract**

L14: also Cenozoic rocks.
L14: Ok. We added Eocene to show the range of stratigraphic ages explored in our study.

L20: "lower plate sediments", do you mean sediments deposited in a basement that subducted, accreted sediments from the subducted plate? Please, clarify.
L20: Modified. We now specify that we are dealing sediments accreted from the downgoing plate.

**Introduction**

I find the introduction a bit short and that fails in emphasizing the relevance of this study to unravel the processes governing the accretion of sediments during collision in the SW Alps. The authors are referring frequently about the subduction processes and the accretion of sediments to the upper plate but there is none crustal-scale cross-section that shows the structure they are referring to. I also find the map of figure 1a too simplified.

The introduction does not aim to present the geology of the Alps, including a lithospheric-scale section of the Alps. We think the section shown in Fig. 1D presents most of the informations on the structure we need in introduction (main thrusts, tectonic units and strtigraphy) and the geological context. A precise description of the Alps is not relevant in the introduction but it can be found in the geological context.

The general description of the Alps and Digne Nappe is insufficient to understand the tectonic context the authors are studying. In this section I would also rather read about an overview of the geology of the western Alps and the rifting episodes with a reference to figures 1a-1c. A proper definition of the Valaisan and Vocontian domains would be useful, as the authors are often referring to them but are not properly described and are relevant to understand fig 6. Regarding the Digne Nappe, it should also be properly described in the geological setting.

We reorganised and included a more lengthy presentation of the geological and tectonic evolution. These modifications are also based on comments by RC1.

It is not very clear what are the previous AFT studies considered in the study. Caption of figure 1b states that they are specified along the text, but until the "samples and method" section, there are no references. I suggest the authors include previous thermochronological studies in the caption of fig. 1, in the introduction as state-of-the-art studies, and maybe also add other reference such as Bogdanoff et al. (2000).

We added the references for thermochronological data in the caption of Fig. 1. We did not add the reference to Bogdanoff et al. (2000) because it is not relevant. Here, we are interested in providing constraints on the eroded cenozoic cover above the Digne Nappe not to present the timing of exhumation in the External Massifs which are all reset for AFT.

**Geological Setting**

This section needs to be reorganized. I would prefer to read first the description of the geology of the study area (crystalline massifs, nappes...) and after that, the tectonic evolution since the Jurassic. In addition, to support the 3 scenarios proposed for the numerical modelling, a more detailed explanation of the geology is needed, including a description of the cross-section of figure 1d.

We added a brief description of the different units before describing the geological history of the Alps.

L67: "Alpine arc", this term has neither been used before nor later, can it be changed to "Alps"? L65-68: I suggest rewriting this sentence: "The second, Late Jurassic-Early Cretaceous in age, appears synchronous with the rifting of the Bay of Biscay and led to opening the Valaisan domain to the NE of the Alpine arc and renewed extension in the so-called Vocontian Trough of the SW Alps" to avoid confusion about the extensional phases affecting the NE and the SW Alps. In addition, is it relevant to mention the rifting of the Bay of Biscay here? In case it is, you need to provide references.

L 67 and 65-68: the sentences have been modified as suggested by RC3 and other comments (RC1 and RC2).

L83: "Helminthoid flysch" is only used here, is this relevant for the study? If not, I suggest to remove it. "sub-Briançonnais" does not appear in Fig. 1a, 1b, only Briançonnais, what do you mean with "sub"?
L 83: We decided to let the word Helminthoid Flysch because it a so-called name for these formations known by every geologist familiar with Alpine geology.
Regarding the use sub-Briançonnais it means units very close to the Briançonnais units. It is also a so-called name of tectonic units in the Alps. It was mentioned in the caption of Fig.  but we added another occurrence in the caption for more clarity.

**Samples and Methods**

L98: Add the number of samples in the beginning of the section.
L 98: modified as suggested.

L99: replace "in rare occasions" for something like: "additionally, two samples were collected in Eocene limestone".
L 99: modified as suggested.

L110: "... of the existing low-temperature fission-track analyses...", add apatite before fission-track.
L 110: modified as suggested.

**Numerical modelling with basin model**

I would rather read first about the software used for the modelling and the parameters chosen, than the scenarios tested. As I am not a user of this software, a brief description of the input parameters, processing and outputs would be very useful.

L127-129: - "Mirabeau well" add the reference to figure 1b.
L 127-129: modified as suggested.

L142-143: "crustal basement with homogeneous properties", in base of what? A continental crust could be very heterogeneous in terms of     lithologies, and therefore, feature different density values. A crystalline crust could imply densities ranging from 2.7 to 3.4 g/cm$^3$ (e.g., Barton, 1986; Rudnick and Fountain, 1995). Other type of models assumes a homogeneous crustal density that increases with depth between 2.6 to 3 g/cm$^3$ (Torne et al., 2015). At lease, provide a reference that supports the value chosen.
For table 1 provide also references and justifications for the values chosen.

The thermal constants chosen for the sedimentary cover are typical values used in basin modelling approach, and they are those already implemented in TemisFlow. We chose to provide details of thermal parameters in the sedimentary cover because they correspond to the lithologies found along our stratigraphic sections. In contrast, the basement is treated as homogeneous with a granitic composition. This is obviously an approximation that is justified by the lithospheric scale.

**Results**

L167: define the acronym AFT before. In addition, add a reference of the AFT study in this sentence.
Done.

L169: Add the reference for the geothermal gradient assumption. For instance, Bigot-Cormier et al. (2006) considered a present geothermal gradient of 25-30 °C/km, and Valla et al. (2011) considered a gradient of 25 °C/km.
L 169: We have added refs to these studies. We note that these estimated gradients have their own limitations.

L181: replace to something like: "the comparison between the temperatures derived from RSCM data and from the numerical modelling…".
L 181: modified as suggested.

L185-186: fig. 3 shows the results for the 3 scenarios, and figure 4 shows the two-rift scenario and one site for the no-rift scenario. Please, be precise.
L 195-186: modified for avoiding misunderstanding.

L205-209: the CAS section is here explained as only burial during the Cenozoic, but how does this fit with the tectonic framework of the rest of the sections which are also located in the Digne Nappe and closer to it (e.g., DGN and CLN)? Maybe a cross-section along this area will help to understand the structure and discuss it more thoroughly…
L 205-209: Agreed. As also suggested by RC1 and in order to keep the interpretation as consistent as possible between nearby samples, we modified the discussion on the CAS section.

L211: "affecting the samples in nature", what do you mean?
We rephrased this part of the ms.

**Discussion**

Here I miss some discussion about the CAS section and why it is assumed to have experienced a constant geothermal gradient of 30ºC/km if it is also located in a rifted margin. I would like to have some more information/discussion about the paleogeographic reconstruction of fig. 6 regarding the location of the rift axis, and transfer zones. It could also help to include in fig. 6a

the location of the study sections. I am aware that they are already included in fig. 6b, but its tentative position in the map would give the reader a better spatial location of the dataset. In addition, a more detailed description of fig. 6b is needed. That would also help to address my previous comment on the explanation of section CAS.

Interpretation of the CAS has been modified and we agree with the reviewer that CAS thermal record also fit with CAS being positioned on the rifted margin. We added the reconstructed positions of the sections along with the restored Digne Nappe in Fig. 6A.

L232. Geothermal gradients of 80-90 ºC/km in the Pyrenees led to high-temperature metamorphism (Ducoux et al., 2021, a reference that the authors cite), and it is accompanied by mantle exhumation to the base of the sedimentary basin or even to the seafloor (e.g., Lagabrielle et al., 2010; DeFelipe et al., 2017, 2019; Teixell et al., 2018).
In addition to Vacherat et al. 2024, only a few studies actually esitmated geothermal gradients (Hart et al., 2017 and Saspiturry et al. 2020). We however cite Ducoux et al. 2021 for reference to HT metamorphism.

The rift domains defined there, and the role of transfer zones are topics that are being highly discussed. If you want to make a proper comparison with the tectonic setting of the Pyrenees (and Bay of Biscay as they also mention it without any reference in L66), you need to add more refences and discuss all these topics.
We do not intend to make a comparison with the Pyrenees or to discuss the issue of transfer zones. We just mention here the Pyrenees because the system is pretty well understood.

Therefore, in your study area, how was the rift system? Which rift domains are described? Is there any evidence of mantle exhumation? To the east of their study area, ophiolite complexes include serpentinized peridotites with ophicalcite (e.g., Lafay et al., 2017).
Our models show that we don't need crustal break up and mantle exhumation to reproduce our data.

In Figure 6a, thinned continental crust is divided into "thinner" and "thicker", can you provide a thickness estimate? Can you also indicate this in fig. 6b? How is it related to the domains of a rifted margin? (e.g., Tugend et al., 2015, a reference that the authors cite).

These informations are already provided in the text.

Figure 6 can also be enlarged.

XXXXX

**Conclusions**
The authors would better summarize their main results here: samples collected, paleotemperature data, scenarios modelled and chosen as representative, and geothermal gradient estimations.

**igures:**

Figure 1: please reorganize the figure to have image "1a" in the top left part of the figure.

1b: - What do broken red lines indicate?

- Please, change the colour of the stratigraphic sections (DEV, SLC, ...). They have the same colour as the reconstructed isopachs.

- Add the definition of the acronym AFT for apatite fission track in the caption.

We corrected the figure and provide details that were lacking.

Figure 2: rewrite the caption. Suggestion: "stratigraphic sections along the front of the Digne Nappe with the RSCM-derived peak temperatures".

Done.

Figure 3: define the tectonic models as two-rifts, one-rift, no-rift along the text to homogenize terms. Place this figure after it is first called (so after L180). Check spelling of color/colour in the caption.

Done.

Figure 4: change the tables of each diagram to something like: "thickness of the crust" and "thickness of the lithospheric mantle". Otherwise, it looks confusing (thickness vs. time).

Modified as suggested.

Figure 6: in the last two lines of the caption: "dashed lines", can you provide a tentative value for each isotherm colour?

Modified as suggested.

In fig. 6b the stratigraphic sections are projected for reference, but I suggest projecting them also on the map of fig. 6a. In the legend, separate the "cover" box into Jurassic and Cretaceous.

The projection of the sections has been added in the palaeogeographic map, as suggested.

**Other (minor) comments:**

L25: " Where details of basin evolution are lacking high-temperature record...", add a comma after lacking.

Modified.

L55: "This study combines 80 new RSCM measurements...", remove the word new.

Modified as suggested.

L79: "currently running between the variscan…", remove currently.

Along the text, the Digne Nappe is referred to with different names: "front of the Digne Nappe", "Digne thrust front", "Digne frontal thrust", "Digne thrust sheet", and "Digne main thrust", please, unify. Please, do the same for "Vocontian Trough", "Vocontian domain", "Vocontian-Valaisan rifting", "Valaisan extensional domain", "Valaisan rift", "Valaisan domain".

We homogeneised through the text.

L198: "the Early-Middle Jurassic" add the "y".

Done.

L221: add **geothermal** before "gradients are about 80-90ºC/km".

Done through the whole text.

L225-226: from where does the β-factor comes from? Literature or your own modelling?

They are calculated from our models values. We added a precision in the text for more clarity.

L276: remove the in "between the Europe and Adria".

Modified.

---

## Author Response (AR2)

Please find in this document the answers to the topical editor comments highlighted in red. Modifications in the manuscript are highlighted in blue.

Dear Authors,
after reading the revised manuscript and your responses to the reviewers' comments, I have concluded that an additional round of revision is required.
Below the four points you need to address:

1) You haven't provided any response for a major point raised during the first round of revision. I'm reporting it below and I strongly encourage you to address this issue in the manuscript, not only in the response letter.
You found very high Temperatures, up to 340°, in the Early-Middle Jurassic beds of the most of the sections. These temperatures are commonly considered to be in the range of metamorphism. Low grade metamorphism (green schist facies), but metamorphism. In the Discussion below you propose an interpretation in which such thermal perturbation is related to two rifting events: this means that such perturbation lasted for tens of million of years. I would expect that sedimentary rocks affected by temperatures above 300°C for such a long time interval in an extensional setting (in which the fluid circulation through the crust is highly favoured) would be transformed in metamorphic rocks. At least they should bear evidence of recrystallization and neo-blastesis. And I guess that many of your samples are shales or rocks with a shaly component, that is the most reactive to metamorphic reactions. You should face this point by providing a petrographic description of your samples. In the case you can not find any evidence of recrystallization and neo-blastesis you should propose a mechanism that hampered these processes at such high temperatures.

First of all, remember that among the 63 samples analysed only two RSCM temperatures yielded temperatures between 300°C and 350°C. As indicated in the revised manuscript most studies in the region argue for the absence in the field of metamorphic fluids. In addition, without sufficient fluid pressures no typical greenschist metamorphism are expected (here pressures were close to 0.9-1.2 kbar below lower greenschist metamorphism at time of peak temperature). Actually, our temperatures are consistent with the illite crystallinity index (Aprahamian, 1988) measured at the base of the Digne Nappe that indicates very low-grade metamorphism in the anchizone. Other minerals like chlorite (Artru, 1967; Levert, 199) have been described but they indicate the rearrangement of clay minerals during diagnesis, not greenschist metamorphism.

I am not an analyst or a Raman expert and I do not want to doubt about the RSCM method; but I know that there has been (and probably there is) debate around it. Some authors reported kinetic effects and the occurrence of metastable poorly crystallized graphitic carbon that, in their opinion, would affect the reliability of the RSCM geothermometer (see for example Foustoukos 2012, American Mineralogist). All the more reason you should describe the rocks you analysed, showing the differences between the ones in the upper part of the sections which not experienced high temperatures and the ones in the lower part which were affected by temperatures up to 340°C. Alternatively, if you could not find any evidence, you should discuss how it was possible, in order to exclude that such temperatures are "fake" temperatures due to analytical artefacts

The kinetic effect is reduced in our samples because the temperatures conditions lasted long enough (several millions of years) so that all our samples reached equilibrium. But most importantly the origin of the CM, which can be distinguished based on the Raman spectra, is very different from other studies studying the effect of organic matter crystallisation associated with fluids. Here we have selected spectra indicating that the organic matter was originally deposited in the sediments, and not originated from metamorphic fluids which could be occasionally present but not observed here. The experiments by Foustoukos (2012) were made at HP-HT conditions which are not met in our studied examples. Our results are consistent with the results obtained from previous studies on illite crystallinity and clay mineralogy. We have reformulated this part of the discussion to strengthen our demonstration.

2) In your work you report the occurrence of a temperature shift of about 100° across the Middle/Upper Jurassic boundary. This shift occurs between samples processed with different calibrations (Lahfid et al. (2010) for temperatures ranging between 200 and 340°C, and Saspiturry et al. (2020) for lower temperatures between 100 and 180°C). Can you rule out that this shift is due to a methodological issue? Having an overlap area (let's say 150-250°) in which samples are processed with both calibrations would greatly help to convince the reader.

This shift cannot be due to a methodological issue because prior to applying one calibration or another, a qualitative analysis of Raman spectra is systematically performed in order to apprehend the range of temperature in which the sample is located. Indeed, the Raman spectra of different temperature ranges strongly vary (see Saspiturry et al., 2020; Lahfid et al., 2010 for examples of Raman spectra evolution with temperature). Thus when we identify the probable temperature range of a sample we apply the relevant calibration. On the contrary it is not relevant to apply two different calibrations on a sample from this "overlap area" because the different calibrations imply the use of different parameters and this would imply strong variations on the final temperature value obtained.

3) Again concerning the shift. If confirmed, such a remarkable temperature shift intuitively requires a thermal event at the end of the Middle Jurassic, rather than during the Cretaceous. Comparison between models and data, indeed, shows that the two-rifting scenario fails to reproduce the abrupt temperature change for 4 of the 6 analyzed sections.

It is noteworthy that ALL the scenarios fail to reproduce the abrupt temperature shift along the sections. This shift is really difficult to obtain by modelling. However, the only scenario that is able to reproduce the highest temperatures at the base of the sections is the Two Rifts-model. Indeed, in the One Rift-model, with the same amount of lithospheric stretching, the temperatures at the base of the sections are not reproduced. Thus, the One Rift-model cannot account for these temperatures.

We agree that the results intuitively point a Middle Jurassic thermal event. However, it would be difficult during the Middle Jurassic to obtain such high temperatures because most of the samples of that age were near the surface of the seafloor where the temperature was close to 0°C. A substantial column of sediments is needed above the Middle Jurassic to have a blanketing effect and to allow the emplacement of a geothermal gradient along the whole section that will allow the Middle Jurassic rocks to

record such high temperatures. This condition is possible during the Early Cretaceous and not during the Middle Jurassic.

We also want to remind that the modelling part has been used to find the scenario which best reproduces the temperatures. We believe that the modelling results are not supposed to reproduce exactly the RSCM data. Rather they should allow us to choose between several scenarios which one allows the closest reproduction of the RSCM temperatures in order to decipher which one is the probable scenario. Looking at the comparison between the RSCM temperatures and the temperatures generated in the models, the Two Rifts-model represent unequivocally the best hypothesis.

4) The Cretaceous tectonic framework of the SE France basin is slightly different from what you are reporting in figure 6. The Vocontian basin, the durance Isthmus, and the South Provence basin form E-W elongated horst and graben structures and, fully in agreement, mid Cretaceous faults in the area point to N-S extension. Crustal scale mid Cretaceous faults in the Brianconnais domain also point to N-S extension, rather than NW-SE oriented extension as you infer.

We agree that the Vocontian domain is EW elongated as well as the Provence high. Our reconstruction does not disagree with this. However, it is true that the Briançonnais should be rotated so to appear more EW directed during the Cretaceous as noticed in previous publications e.g. Angrand and Mouthereau (2021); Lemoine et al. (1986), Dumont al. (2012), or Beltrando et al. (2012). We modified figure 6 to keep the northern border of the Briançonnais aligned with Provence in a more EW direction.

The above points should be solved or at least you should add a section in the discussion, in which you list the limitations/discrepancies of your model

In summary, we think that the only limitations are related to the use of 1D thermal approach for solving a problem which is 3D. This is clearly stated in the paper. However, we reiterate that the RSCM results are not biased. The limitations raised by other studies (kinetics, fluids, deformation) on the CM geothermometry simply do not apply here.

---

## Author Response (AR3)

We have made substantial changes of the original version of our paper submitted to Short Communication of EGU Solid Earth publications thanks to comments by 3 reviewers and during a second round of reviews motivated by new comments made by the topical editor. We are here responding to a third round of reviews following comments by the Topical Editor S. Tavani.

Point 1 arises from the curiosity of the first reviewer about your temperatures of 300° obtained in a sedimentary sequence that is commonly considered by geologist working in that area as non metamorphic. I share this curiosity and I suspect that many readers will do the same. I recognize your effort in providing a response to this question but it is not fully clear why you do not want to add petrographic description of the "hotter" samples, as suggested by the reviewer #1. I would really appreciate having in the supplementary material images of thin sections of the samples for which you have obtained temperatures around 300°.

Unfortunately, we have not taken photographs of the carbonate samples while performing the Raman analysis. We understand that this might be curious for the readers. But the reason is that the temperatures are obtained on organic matter that did not originate from metamorphic fluids. This was controlled by looking at the Raman spectra itself. Moreover, the maturation state of the organic matter is not visible on the microscope, and it is not dependent on the presence or not of newly formed minerals (which was not seen as we were analyzing the samples). We thus think that such images will not help the reader.
Moreover, as we described in the previous minor review round, the lack of metamorphism is well-known and for a long time by the geologists working in the region and this has been substantiated by a number of papers that we cited. We have proposed some directions to explain why there is no metamorphism in the revised version (lack of significant metamorphic fluid flow and low pressures) but exploring why there is no metamorphism in the Digne Nappe is not the goal of our paper.

Point 2 As the authors know, Raman derived temperatures in diagenetic environments are problematic, and the application of the qualitative method by Saspiturry et al. (2020) could be considered questionable by many readers. Moreover, the fact that your thermal jump occurs at the transition between temperatures estimated by two different methods (one of which is by Saspiturry et al.), strongly calls the attention. In order to demonstrate the occurrence of the temperature jump, if you do not want to use the two methods in an overlap area, in figure 2 you should at least add the graphs for some raman spectra parameters, such as the RBS (Raman band separation) or the R1 (intensity ratio between D and G).

We agree that the reader must be able to see directly in the article what the Raman spectra look like. We have integrated a new figure (fig. 2 in the new version of the manuscript) where we present representative Raman spectra for samples from this study, with the RSCM temperatures obtained and the R1 and RBS parameters as the Editor suggested. In addition, we provide a supplementary file presenting the position and intensity values of the G and D bands and RBS and R1 parameters of the spectra shown in Fig. 2.

Point 3
Com'on! If you tune properly the parameters a polynomial function of degree N compared to a polynomial function of degree N-1 will always provide a better fit. So, if you add a third rifting event your RMS will surely decrease. The point is that you should make crystal clear to the reader that despite you are adding a second rifting event (i.e. despite you are increasing the

degree of the polynomial function) the temperature shift across Middle-Late Jurassic boundary still cannot be modeled.

Perhaps there is some misunderstanding here. No interpolation was made in new Figure 4 (Fig.3 old). The RMS evaluates the goodness of fit between observed (circles) and calculated maximum temperatures (diamond, square and triangle symbols) extracted from our thermal modelling. The problem is non-linear. Calculate Tmax depend on when and how rift-related heating (tectonics) occurred relative to burial heating (sediment burial). For instance, adding a third rifting event after the second Cretaceous one would not change the result. This is because of the limited post-Cretaceous sediment burial and because raise temperatures above the values reached in the early Cretaceous would require to drastically increase lithospheric thinning after the early Cretaceous which makes no sense in the geological context. Also it must be remembered that the temperature-depth relationships presented in new figure 4 are apparent geothermal gradients, this is the reason why we also present in new Figure 6 the evolution geothermal gradients recorded in the stratigraphic column. So to sum up, yes it is obvious that the shift cannot be reproduced because the temperatures above and below are reached during two different periods (early Cretaceous and Cenozoic) and tectonic settings (rifting and foreland basin). The model we propose has the advantage to reproduce 1) the maximum temperatures of around 300°C that can only be achieved in the early Cretaceous when burial and thinning act together, and reproduce at the same time 2) the low-temperature in the upper samples and the temperature above 200°C in the lower samples. Note that none of the three reviewers have raised any issues concerning the modelling approach.

Point 4
I understand that in this paper you do not want to discuss about the connection between the NE-SW striking Valaisian structures and the WNW-ESE striking "Pyrenean" structures of the SE France basin/Brianconnais. However, your figure 6 suggests the two sets of structures form part of the same framework. Either you tone down such a graphic suggestion (i.e. removing the faults in Vocontian basin and the entire Provence platform from your figure 6) or you discuss the bigger picture and explicate you view on how the Pyrenean and Valesian rifts connects.

Why should we remove the faults in the Vocontian basin and the Provence ? First this reconstruction that implies a connection between the Valais Basin and the Vocontian Basin has been proposed in many other studies (that we referred to) and as you said we don't want to elaborate more on the connection between the Valais Basin and the Pyrenean rift (not shown here). Again, none of the reviewers commented on this reconstruction and we already modified it based on your previous suggestion.

Naïm Célini and co-authors.